# Joint processing of linguistic properties in brains and language models

**Subba Reddy Oota**[1,2]**, Manish Gupta**[3]**, Mariya Toneva**[2]

[1]Inria Bordeaux, France, [2]MPI for Software Systems, Saarbrücken, Germany, [3]Microsoft, India

`subba-reddy.oota@inria.fr, gmanish@microsoft.com, mtoneva@mpi-sws.org`

## Abstract

Language models have been shown to be very effective in predicting brain recordings of subjects experiencing complex language stimuli. For a deeper understanding of this alignment, it is important to understand the correspondence between the detailed processing of linguistic information by the human brain versus language models. We investigate this correspondence via a direct approach, in which we eliminate information related to specific linguistic properties in the language model representations and observe how this intervention affects the alignment with fMRI brain recordings obtained while participants listened to a story. We investigate a range of linguistic properties (surface, syntactic, and semantic) and find that the elimination of each one results in a significant decrease in brain alignment. Specifically, we find that syntactic properties (i.e. Top Constituents and Tree Depth) have the largest effect on the trend of brain alignment across model layers. These findings provide clear evidence for the role of specific linguistic information in the alignment between brain and language models, and open new avenues for mapping the joint information processing in both systems. We make the code publicly available[1].

## 1 Introduction

Language models that have been pretrained for the next word prediction task using millions of text documents can significantly predict brain recordings of people comprehending language (Wehbe et al., 2014; Jain & Huth, 2018; Toneva & Wehbe, 2019; Caucheteux & King, 2020; Schrimpf et al., 2021; Goldstein et al., 2022). Understanding the reasons behind the observed similarities between language comprehension in machines and brains can lead to more insight into both systems.

While such similarities have been observed at a coarse level, it is not yet clear whether and how the two systems align in their information processing pipeline. This pipeline has been studied separately in both systems. In natural language processing (NLP), researchers use probing tasks to uncover the parts of the model that encode specific linguistic properties (e.g. sentence length, tree depth, top constituents, tense, bigram shift, subject number, object number) (Adi et al., 2016; Hupkes et al., 2018; Conneau et al., 2018; Jawahar et al., 2019; Rogers et al., 2020). These techniques have revealed a hierarchy of information processing in multi-layered language models that progresses from simple to complex with increased depth. In cognitive neuroscience, traditional experiments study the processing of specific linguistic properties by carefully controlling the experimental stimulus and observing the locations or time points of processing in the brain that are affected the most by the controlled stimulus (Hauk & Pulvermüller, 2004; Pallier et al., 2011).

More recently, researchers have begun to study the alignment of these brain language regions with the layers of language models, and found that the best alignment was achieved in the middle layers of these models (Jain & Huth, 2018; Toneva & Wehbe, 2019; Caucheteux & King, 2020). This has been

---

[1]`https://github.com/subbareddy248/lingprop-brain-alignment`

37th Conference on Neural Information Processing Systems (NeurIPS 2023).

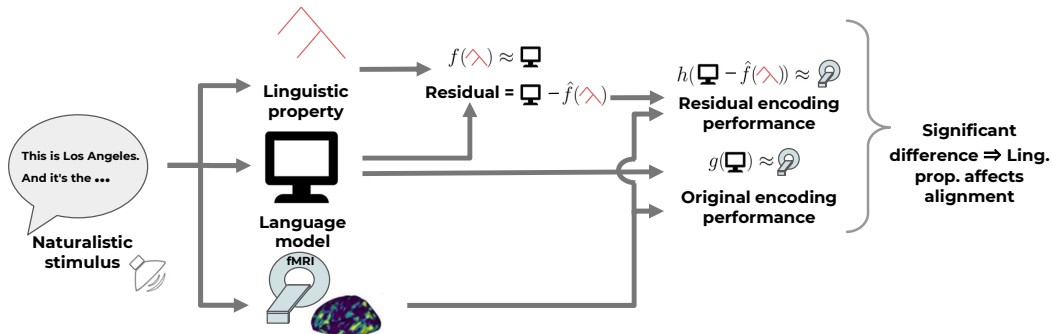

Figure 1: Approach to directly test for the effect of a linguistic property on the alignment between a language model and brain recordings. First, we remove the linguistic property from the language model representations by learning a simple linear function $f$ that maps the linguistic property to the language model representations, and use this estimated function to obtain the residual language model representation without the contribution of the linguistic property. Next, we compare the brain alignment (i.e. encoding model performance) before and after the removal of the linguistic property by learning simple linear functions $g$ and $h$ that map the full and residual language model representations to the brain recordings elicited by the corresponding words. Finally, we test whether the differences in brain alignment before and after the removal of the linguistic property are significant and, if so, conclude that the respective linguistic property affects the alignment between the language model and brain recordings.

hypothesized to be because the middle layers may contain the most high-level language information as they are farthest from the input and output layers, which contain word-level information due to the self-supervised training objective. However, this hypothesis is difficult to reconcile with the results from more recent NLP probing tasks Conneau et al. (2018); Jawahar et al. (2019), which suggest that the deepest layers in the model should represent the highest-level language information. Taken together, these findings open up the question of what linguistic information underlies the observed alignment between brains and language models.

Our work aims to examine this question via a direct approach (see Figure 1 for a schematic). For a number of linguistic properties, we analyze how the alignment between brain recordings and language model representations is affected by the elimination of information related to each linguistic property. For the purposes of this work, we focus on one popular language model–BERT (Devlin et al., 2018)–which has both been studied extensively in the NLP interpretability literature (i.e. BERTology Jawahar et al. (2019)) and has been previously shown to significantly predict fMRI recordings of people processing language (Toneva & Wehbe, 2019; Schrimpf et al., 2021). We test the effect of a range of linguistic properties that have been previously shown to be represented in pretrained BERT (Jawahar et al., 2019). We use a dataset of fMRI recordings that are openly available (Nastase et al., 2021) and correspond to 18 participants listening to a natural story.

Using this direct approach, we find that the elimination of each linguistic property results in a significant decrease in brain alignment across all layers of BERT. We additionally find that the syntactic properties (Top Constituents and Tree Depth) have the highest effect on the trend of brain alignment across model layers. Specifically, Top Constituents is responsible for the bump in brain alignment in the middle layers for all language regions whereas Tree Depth has an impact for temporal (ATL and PTL) and frontal language regions (IFG and MFG). Performing the same analyses with a second popular language model, GPT2 (Radford et al., 2019), yielded similar results (see Appendix section F).

Our main contributions are as follows:

1. We propose a direct approach to evaluate the joint processing of linguistic properties in brains and language models.
2. We show that removing specific linguistic properties leads to a significant decrease in brain alignment. We find that the tested syntactic properties are the most responsible for the trend of brain alignment across BERT layers.
3. Detailed region and sub-region analysis reveal that properties that may not impact the whole brain alignment trend, may play a significant role in local trends (e.g., Object Number for

ATL and IFGOrb regions, Tense for PCC regions, Sentence Length and Subject Number for PFm sub-region).

We make the code publicly available[1].

## 2 Related Work

Our work is most closely related to that of Toneva et al. (2022), who employ a similar residual approach to study the supra-word meaning of language by removing the contribution of individual words to brain alignment. We build on this approach to study the effect of specific linguistic properties on brain alignment across layers of a language model. Other recent work has examined individual attention heads in BERT and shown that their performance on syntactic tasks correlate with their brain encoding performance in several brain regions (Kumar et al., 2022). Our work presents a complementary approach that gives direct evidence for the importance of a linguistic property to brain alignment.

This work also relates to previous works that investigated the linguistic properties encoded across the layer hierarchy of language models (Adi et al., 2016; Hupkes et al., 2018; Conneau et al., 2018; Jawahar et al., 2019; Rogers et al., 2020). Unlike these studies in NLP, we focus on understanding the degree to which various linguistic properties impact the performance of language models in predicting brain recordings.

Our work also relates to a growing literature that relates representations of words in language models to those in the brain. A number of studies have related brain responses to word embedding methods (Pereira et al., 2016; Anderson et al., 2017; Pereira et al., 2018; Toneva & Wehbe, 2019; Hollenstein et al., 2019; Wang et al., 2020), sentence representation models (Sun et al., 2019; Toneva & Wehbe, 2019; Sun et al., 2020), recurrent neural networks (Jain & Huth, 2018; Oota et al., 2019), and Transformer methods (Gauthier & Levy, 2019; Toneva & Wehbe, 2019; Schwartz et al., 2019; Jat et al., 2020; Schrimpf et al., 2021; Goldstein et al., 2022; Oota et al., 2022b,a; Merlin & Toneva, 2022; Aw & Toneva, 2023; Oota et al., 2023). Our approach is complementary to these previous works and can be used to further understand the reasons behind the observed brain alignment.

## 3 Dataset Curation

**Brain Imaging Dataset**    The "Narratives" collection aggregates a variety of fMRI datasets collected while human subjects listened to naturalistic spoken stories. We analyze the "Narratives-21st year" dataset (Nastase et al., 2021), which is one of the largest publicly available fMRI datasets (in terms of number of samples per participant). The dataset contains data from 18 subjects who listened to the story titled "21st year". The dataset for each subject contains 2226 samples (TRs-Time Repetition). The dataset was already preprocessed and projected on the surface space ("fsaverage6"). We use the multi-modal parcellation of the human cerebral cortex (Glasser Atlas: consists of 180 ROIs in each hemisphere) to display the brain maps (Glasser et al., 2016), since the Narratives dataset contains annotations that correspond to this atlas. The data covers seven language brain regions of interest (ROIs) in the human brain with the following subdivisions: (i) angular gyrus (AG: PFm, PGs, PGi, TPOJ2, and TPOJ3); (ii) anterior temporal lobe (ATL: STSda, STSva, STGa, TE1a, TE2a, TGv, and TGd); (iii) posterior temporal lobe (PTL: A4, A5, STSdp, STSvp, PSL, STV, TPOJ1); (iv) inferior frontal gyrus (IFG: 44, 45, IFJa, IFSp); (v) middle frontal gyrus (MFG: 55b); (vi) inferior frontal gyrus orbital (IFGOrb: a47r, p47r, a9-46v), (vii) posterior cingulate cortex (PCC: 31pv, 31pd, PCV, 7m, 23, RSC); and (viii) dorsal medial prefrontal cortex (dmPFC: 9m, 10d, d32) (Baker et al., 2018; Milton et al., 2021; Desai et al., 2022).

**Extracting Word Features from BERT**    We investigate the alignment with the base pretrained BERT model, provided by Hugging Face (Wolf et al., 2020) (12 layers, 768 dimensions). The primary rationale behind our choice to analyze the BERT model comes from the extensive body of research that aims to delve into the extent to which diverse English language structures are embedded within Transformer-based encoders, a field often referred to as "BERTology" (Jawahar et al., 2019; Mohebbi et al., 2021). We follow previous work to extract the hidden-state representations from each layer of pretrained BERT, given a fixed-length input (Toneva & Wehbe, 2019). To extract the stimulus features from pretrained BERT, we constrained the tokenizer to use a maximum context of previous

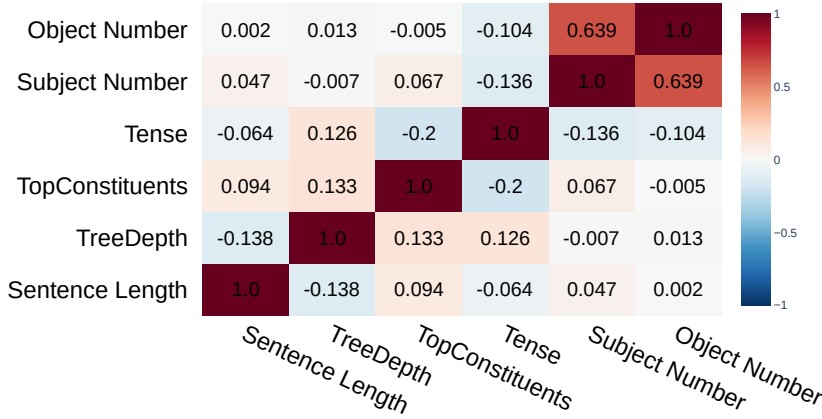

Figure 2: Task Similarity (Pearson Correlation Coefficient) constructed from the task-wise labels across six tasks. We observe a high correlation only between Subject Number and Object Number. There are a number of small positive and negative correlations among the remaining properties.

20 words. Given the constrained context length, each word is successively input to the network with at most $C$ previous tokens. For instance, given a story of $M$ words and considering the context length of 20, while the third word's vector is computed by inputting the network with $(w_1, w_2, w_3)$, the last word's vectors $w_M$ is computed by inputting the network with $(w_{M-20}, \ldots, w_M)$. The pretrained Transformer model outputs word representations at different encoder layers. We use the #words × 768 dimension vector obtained from each hidden layer to obtain word-level representations from BERT. For the analyses using GPT-2, we extract the stimulus features in an identical way from a pretrained GPT-2, provided by Hugging Face (Wolf et al., 2020) (12 layers, 768 dimensions).

**Downsampling**    Since the rate of fMRI data acquisition (TR = 1.5sec) was lower than the rate at which the stimulus was presented to the subjects, several words fall under the same TR in a single acquisition. Hence, we match the stimulus acquisition rate to fMRI data recording by downsampling the stimulus features using a 3-lobed Lanczos filter. After downsampling, we obtain the chunk-embedding corresponding to each TR.

**TR Alignment**    To account for the delay of the hemodynamic response, we model the hemodynamic response function using a finite response filter (FIR) per voxel and for each subject separately with 8 temporal delays corresponding to 12 seconds (Nishimoto et al., 2011).

**Probing Tasks**    Probing tasks (Adi et al., 2016; Hupkes et al., 2018; Jawahar et al., 2019) help in unpacking the linguistic features possibly encoded in neural language models. In this paper, we use six popular probing tasks grouped into three categories–surface (sentence length), syntactic (tree depth and top constituents), and semantic (tense, subject number, and object number)–to assess the capability of BERT layers in encoding each linguistic property. Since other popular probing tasks such as Bigram Shift, Odd-Man-Out, and Coordination Inversion (as discussed in Conneau et al. (2018)) have constant labels for our brain dataset, we focused on the remaining six probing tasks mentioned above. Details of each probing task are as follows. **Sentence Length:** This surface task tests for the length of a sentence. **TreeDepth:** This syntactic task tests for classification where the goal is to predict depth of the sentence's syntactic tree (with values ranging from 5 to 12). **TopConstituents:** This syntactic task tests for the sequence of top-level constituents in the syntactic tree. This is a 20-class classification task (e.g. ADVP_NP_NP_, CC_ADVP_NP_VP_). **Tense:** This semantic task tests for the binary classification, based on whether the main verb of the sentence is marked as being in the present (PRES class) or past (PAST class) tense. **Subject Number:** This semantic task tests for the binary classification focusing on the number of the subject of the main clause of a sentence. The classes are NN (singular) and NNS (plural or mass: "colors", "waves", etc). **Object Number:** This semantic task tests for the binary classification focusing on the object number in the main clause of a sentence. Details of these tasks are mentioned in Conneau et al. (2018).

**Balancing Classes**   As discussed in Section 4, our method involves removing one linguistic property $p$ at a time. Ideally, removing $p$ should not impact the performance for any other property $p'$. However, if some classes of $p$ co-occur very frequently with a few classes of $p'$, this undesired behavior may happen. The Left column of Appendix Figure 6 shows the presence of such frequent class co-occurrences across properties. To avoid skewed class co-occurrences, we regroup the class labels for three tasks: Sentence Length, TreeDepth and TopConstituents, as shown in Appendix Figure 5.

**Probing Task Annotations**   To annotate the linguistic property labels for each probing task for our dataset, we use Stanford core-NLP stanza library (Manning et al., 2014) for sentence-level annotations. Further, for word-level annotations, we assign the same label for all the words present in a sentence except for the sentence length task.

**Probing Tasks Similarity**   Pearson correlation values between task labels for each pair of tasks were used to construct the similarity matrix for the "21st year" stimulus text, as shown in Figure 2. We observe a high correlation only between Subject Number and Object Number. There are a number of small positive and negative correlations among the remaining properties.

## 4   Methodology

**Removal of Linguistic Properties**   To remove a linguistic property from the pretrained BERT representations, we use a ridge regression method in which the probing task label is considered as input and the word features are the target. We compute the residuals by subtracting the predicted feature representations from the actual features resulting in the (linear) removal of a linguistic property from pretrained features (see Figure 1 for a schematic). Because the brain prediction method is also a linear function (see next paragraph), this linear removal limits the contribution of the linguistic property to the eventual brain prediction performance.

Specifically, given an input matrix $\mathbf{T}_i$ with dimension $\mathbf{N} \times 1$ for probing task $i$, and target word representations $\mathbf{W} \in \mathbb{R}^{\mathbf{N} \times \mathbf{d}}$, where $\mathbf{N}$ denotes the number of words (8267) and $\mathbf{d}$ denotes the dimensionality of each word (768 dimension), the ridge regression objective function is $f(\mathbf{T}_i) = \min_{\theta_i} \|\mathbf{W} - \mathbf{T}_i \theta_i\|_F^2 + \lambda \|\theta_i\|_F^2$ where $\theta_i$ denotes the learned weight coefficient for embedding dimension $\mathbf{d}$ for the input task $i$, $\|.\|_F^2$ denotes the Frobenius norm, and $\lambda > 0$ is a tunable hyper-parameter representing the regularization weight for each feature dimension. Using the learned weight coefficients, we compute the residuals as follows: $r(\mathbf{T}_i) = \mathbf{W} - \mathbf{T}_i \theta_i$.

We verified that another popular method of removing properties from representations–Iterative Null Space Projection (INLP) (Ravfogel et al., 2020)–leads to similar results (see Appendix Table 4). We present the results from the ridge regression removal in the main paper due to its simplicity.

Similar to the probing experiments with pretrained BERT, we also remove linguistic properties from GPT-2 representations and observe similar results (see Appendix Table 11).

**Voxelwise Encoding Model**   To explore how linguistic properties are encoded in the brain when listening to stories, we use layerwise pretrained BERT features as well as residuals by removing each linguistic property and using them in a voxelwise encoding model to predict brain responses. If a linguistic property is a good predictor of a specific brain region, information about that property is likely encoded in that region. In this paper, we train fMRI encoding models using Banded ridge regression (Tikhonov et al., 1977) on stimulus representations from the feature spaces mentioned above. Before doing regression, we first z-scored each feature channel separately for training and testing. This was done to match the features to the fMRI responses, which were also z-scored for training and testing. The solution to the banded regression approach is given by $f(\hat{\beta}) = \underset{\beta}{\mathrm{argmin}} \|\mathbf{Y} - \mathbf{X}\beta\|_F^2 + \lambda \|\beta\|_F^2$, where $\mathbf{Y}$ denotes the voxels matrix across TRs, $\beta$ denotes the learned regression coefficients, and $\mathbf{X}$ denotes stimulus or residual representations. To find the optimal regularization parameter for each feature space, we use a range of regularization parameters that is explored using cross-validation. The main goal of each fMRI encoding model is to predict brain responses associated with each brain voxel given a stimulus.

Table 1: Word-Level Probing task performance for each BERT layer before and after removal of each linguistic property using the $21^{st}$ year stimuli.

| Layers | Sentence Length 3-classes (Surface) | | TreeDepth 3-classes (Syntactic) | | TopConstituents 2-classes (Syntactic) | | Tense 2-classes (Semantic) | | Subject Number 2-classes (Semantic) | | Object Number 2-classes (Semantic) | |
|---|---|---|---|---|---|---|---|---|---|---|---|---|
| | before | after | before | after | before | after | before | after | before | after | before | after |
| 1 | **74.67** | 43.28 | 76.30 | 42.93 | 77.15 | 47.28 | 87.00 | 59.25 | 92.10 | 49.95 | 93.28 | 47.31 |
| 2 | 69.83 | 42.44 | 76.72 | 38.88 | 78.60 | 42.75 | 87.18 | 48.25 | 92.32 | 55.50 | 93.47 | 54.59 |
| 3 | 72.31 | 46.19 | 75.76 | 40.33 | 77.81 | 48.85 | 87.42 | 44.26 | 93.04 | 48.55 | 93.80 | 49.76 |
| 4 | 71.34 | 46.43 | 75.94 | 38.63 | 78.36 | 48.00 | 88.09 | 42.56 | 93.50 | 50.12 | 94.90 | 50.06 |
| 5 | 72.67 | 46.97 | 76.00 | 40.88 | 78.60 | 45.28 | 88.39 | 44.26 | 94.05 | 49.88 | 93.59 | 51.45 |
| 6 | 70.38 | 44.37 | **79.02** | 41.89 | **80.23** | 43.47 | 87.17 | 44.44 | 94.98 | 55.08 | 94.50 | 54.17 |
| 7 | 72.98 | 46.55 | 77.93 | 41.23 | **80.23** | 46.43 | 88.69 | 42.62 | 95.88 | 50.24 | 94.62 | 47.58 |
| 8 | 72.67 | 44.67 | 76.07 | 40.08 | 78.90 | 46.86 | 87.42 | 44.56 | 96.10 | 50.24 | **95.10** | 50.18 |
| 9 | 70.50 | 45.28 | 77.15 | 42.62 | 79.87 | 44.55 | 88.27 | 47.22 | 96.38 | 52.78 | 94.56 | 49.27 |
| 10 | 72.91 | 47.93 | 76.90 | 41.78 | 78.17 | 47.76 | **88.94** | 45.47 | 96.06 | 53.68 | 94.50 | 50.30 |
| 11 | 70.07 | 46.67 | 77.27 | 45.47 | 77.69 | 45.77 | 87.24 | 48.43 | **96.94** | 53.44 | 94.92 | 49.52 |
| 12 | 71.77 | 42.93 | 76.39 | 46.61 | 78.29 | 48.67 | 86.88 | 45.10 | 94.03 | 51.45 | 93.95 | 48.73 |

**Cross-Validation**   The ridge regression parameters were fit using 4-fold cross-validation. All the data samples from K-1 folds were used for training, and the generalization was tested on samples from the left-out fold.

**Evaluation Metrics**   We evaluate our models using Pearson Correlation (PC) which is a popular metric for evaluating brain alignment (Jain & Huth, 2018; Schrimpf et al., 2021; Goldstein et al., 2022). Let TR be the number of time repetitions. Let $Y = \{Y_i\}_{i=1}^{TR}$ and $\hat{Y} = \{\hat{Y}_i\}_{i=1}^{TR}$ denote the actual and predicted value vectors for a single voxel. Thus, $Y \in R^{TR}$ and also $\hat{Y} \in R^{TR}$. We use Pearson Correlation (PC) which is computed as $corr(Y, \hat{Y})$ where corr is the correlation function.

**Implementation Details for Reproducibility**   All experiments were conducted on a machine with 1 NVIDIA GEFORCE-GTX GPU with 16GB GPU RAM. We used banded ridge-regression with the following parameters: MSE loss function, and L2-decay ($\lambda$) varied from $10^1$ to $10^3$; the best $\lambda$ was chosen by tuning on validation data; the number of cross-validation runs was 4.

# 5   Results

## 5.1   Successful removal of linguistic properties from pretrained BERT

To assess the degree to which different layers of pretrained BERT representation encode labels for each probing task, a probing classifier is trained over it with the embeddings of each layer. For all probing tasks, we use logistic regression as the probing classifier. We train different logistic regression classifiers per layer of BERT using six probing sentence-level datasets created by Conneau et al. (2018). At test time, for each sample from $21^{st}$ year, we extract stimuli representations from each layer of BERT, and perform classification for each of the six probing tasks using the learned logistic regression models. To investigate whether a linguistic property was successfully removed from the pretrained representations, we further test whether we can decode the task labels from the *residuals* for each probing task of the $21^{st}$ year stimuli.

Table 1 reports the result for each probing task, before and after removal of the linguistic property from pretrained BERT. We also report similar results for GPT2 in Table 11 in the Appendix. Similarly to earlier works (Jawahar et al., 2019), we find that BERT embeds a rich hierarchy of linguistic signals also for our $21^{st}$ year stimuli: surface information at the initial to middle layers, syntactic information in the middle, semantic information at the middle to top layers. We verify that the removal of each linguistic property from BERT leads to reduced task performance across all layers, as expected.

We further test whether removing information about one property affects the decoding performance for another property. We observed that most properties are not substantially affected by removing other task properties (see Appendix Tables 5, 6, 7, 8, 9 and 10).

## 5.2   Removal of linguistic properties significantly decreases brain alignment across all layers

In Figure 3 (*left*), we present the average brain alignment across all layers of pretrained BERT before and after the removal of each linguistic property. Removing each linguistic property leads

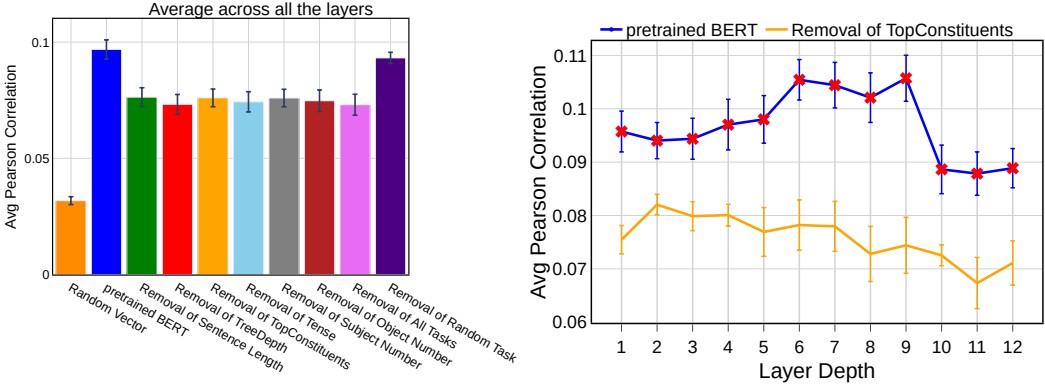

Figure 3: Brain alignment of pretrained BERT before (blue) and after (all other bars) removal of different linguistic properties. *Left* plot compares the average Pearson correlation across all layers of pretrained BERT and all voxels, and the same quantity after removal of each linguistic property. The error bars indicate the standard error of the mean across participants. The *right* plot compares the layer-wise performance of pretrained BERT and removal of one linguistic property–TopConstituents. A red ∗ at a particular layer indicates that the alignment from the pretrained model is significantly reduced by the removal of this linguistic property at this particular layer. The layer-wise results for removing other linguistic properties differ across properties, but are significant at the same layers as TopConstituents and are presented in Appendix Figure 7.

to a significant reduction in brain alignment. In contrast, removing a randomly generated linguistic property vector (for this baseline, think about a new task which has no linguistic meaning. The values for each row is then set to a random value chosen uniformly from across all class labels) does not significantly affect the brain alignment. This validates that the reduction in brain alignment observed after removing the linguistic information is indeed due to the elimination of linguistic information, rather than to the removal method itself. Further, we observe that the brain alignment of a random vector is significantly lower than the residual brain alignment after removal of each linguistic property. This suggests that there are additional important properties for the alignment between pretrained BERT and the fMRI recordings, beyond the ones considered in this work.

In Figure 3 (right), we also report the layer-wise performance for pretrained BERT before and after the removal of one representative linguistic property–TopConstituents. We present the results for the remaining properties in Figure 7 in the Appendix. We also report similar results for GPT2 in Figure11 in the Appendix. Similarly to previous work (Toneva & Wehbe, 2019), we observe that pretrained BERT has the best brain alignment in the middle layers. We observe that the brain alignment is reduced significantly across all layers after the removal of the linguistic property (indicated with red cross in the figure). We observe the most substantial reductions in brain alignment in middle to late layers. This pattern holds across all tested linguistic properties. These results provide direct evidence that these linguistic properties in fact significantly affect the alignment between fMRI recordings and pretrained BERT.

**Statistical Significance**    To estimate the statistical significance of the performance differences (across all tasks), we performed a two-tailed paired-sample t-test on the mean correlation values for the subjects. In all cases, we report p-values across layers. For all layers, the main effect of the two-tailed test was significant for all the tasks with $p \leq 0.05$ with confidence 95% . Finally, the Benjamni-Hochberg False Discovery Rate (FDR) correction for multiple comparisons (Benjamini & Hochberg, 1995) is used for all tests (appropriate because fMRI data is considered to have positive dependence (Genovese, 2000)). Overall, the p-values confirmed that removing each linguistic property from BERT significantly reduced brain alignment.

Figure 10 in the Appendix displays p-values for the brain alignment of pretrained BERT before and after the removal of each linguistic property across all the layers. We observe that the alignment with the whole brain is significantly reduced across all the layers.

**ROI-Level Analysis**    We further examine the effect on the alignment specifically in a set of regions of interest (ROI) that are thought to underlie language comprehension (Fedorenko et al., 2010;

Table 2: ROI-level (*left*) and whole brain (*right*) correlations across layers between 1) differences in decoding task performance before and after removing each linguistic property and 2) differences in brain alignment of pretrained BERT before and after removing the corresponding linguistic property.

| Tasks | AG | ATL | PTL | IFG | IFGOrb | MFG | PCC | dmPFC | Whole Brain |
|---|---|---|---|---|---|---|---|---|---|
| Sentence Length | 0.261 | 0.264 | 0.220 | 0.355 | 0.129 | 0.319 | 0.143 | 0.100 | 0.216 |
| TreeDepth | 0.365 | **0.421** | **0.458** | **0.442** | 0.257 | **0.436** | 0.109 | 0.027 | **0.443** |
| TopConstituents | **0.489** | **0.421** | **0.464** | **0.516** | **0.453** | **0.463** | **0.459** | **0.463** | **0.451** |
| Tense | 0.226 | 0.283 | 0.307 | 0.325 | 0.345 | 0.339 | **0.435** | 0.122 | 0.248 |
| Subject Number | 0.124 | 0.201 | 0.231 | 0.239 | 0.285 | 0.228 | 0.348 | 0.237 | 0.254 |
| Object Number | 0.306 | **0.392** | 0.342 | 0.313 | **0.503** | 0.335 | 0.328 | 0.001 | 0.263 |

Table 3: Finer-grained sub-ROI-level correlations across layers between 1) differences in decoding task performance before and after removing each linguistic property and 2) differences in brain alignment of pretrained BERT before and after removing the corresponding linguistic property.

| Tasks | AG | | | ATL | | PTL | | | | | | IFG | | | |
|---|---|---|---|---|---|---|---|---|---|---|---|---|---|---|---|
| | PFm | PGi | PGs | STGa | STSda | STSdp | A5 | TPOJ1 | PSL | STV | SFL | 44 | 45 | IFJa | IFSp |
| Sentence Length | **0.386** | **0.381** | 0.286 | 0.258 | 0.227 | 0.210 | 0.176 | 0.230 | 0.317 | 0.246 | 0.231 | 0.317 | 0.372 | **0.397** | 0.357 |
| TreeDepth | 0.356 | **0.461** | **0.479** | **0.414** | **0.515** | **0.550** | **0.525** | **0.469** | 0.287 | **0.458** | **0.418** | **0.430** | **0.474** | **0.439** | **0.516** |
| TopConstituents | **0.479** | **0.526** | **0.454** | 0.336 | **0.426** | 0.425 | **0.477** | 0.428 | **0.463** | 0.360 | **0.398** | **0.531** | **0.581** | 0.353 | **0.413** |
| Tense | 0.369 | 0.365 | 0.276 | 0.279 | 0.238 | 0.368 | 0.309 | 0.340 | 0.314 | 0.328 | 0.329 | 0.285 | 0.377 | 0.321 | 0.338 |
| Subject Number | 0.335 | 0.134 | 0.068 | 0.271 | 0.270 | 0.225 | 0.246 | 0.158 | 0.302 | 0.226 | 0.256 | 0.283 | 0.267 | 0.271 | 0.164 |
| Object Number | **0.396** | 0.203 | 0.298 | **0.443** | **0.438** | 0.370 | 0.287 | 0.328 | 0.279 | **0.398** | 0.310 | 0.319 | 0.335 | 0.318 | 0.321 |

Fedorenko & Thompson-Schill, 2014) and word semantics (Binder et al., 2009). We find that across language regions, the alignment is significantly decreased by the removal of the linguistic properties across all layers (see Appendix Figure 8). We further investigate the language sub-regions, such as 44, 45, IFJa, IFSp, STGa, STSdp, STSda, A5, PSL, and STV and find that they also align significantly worse after the removal of the linguistic properties from pretrained BERT (see Appendix Figure 9).

## 5.3 Decoding task performance vs brain alignment

While the previous analyses revealed the importance of linguistic properties for brain alignment at individual layers, we would also like to understand the contribution of each linguistic property to the trend of brain alignment across layers. For this purpose, we perform both quantitative and qualitative analyses by measuring the correlation across layers between the differences in decoding task performance from pretrained and residual BERT and the differences in brain alignment of pretrained BERT and residual BERT. A high correlation for a specific linguistic property suggests a strong relationship between the presence of this property across layers in the model and its corresponding effect on the layer-wise brain alignment. Note that this analysis can provide a more direct and stronger claim for the effect of a linguistic property on the brain alignment across layers than alternate analyses, such as computing the correlation between the decoding task performance and the brain alignment across layers only at the level of the pretrained model. We present the results of these analyses for the whole brain and language regions, and language sub-regions in Tables 2 and 3, respectively.

**Whole Brain Analysis**    High correlations in the last column of Table 2 indicate that the syntactic NLP tasks TopConstituents and TreeDepth have the strongest effect on the alignment between BERT and the whole brain. At the level of the whole brain, surface level and semantic properties have a moderate effect on the trend in brain alignment across layers.

**ROI-Level Analysis**    Further, we analyze these correlations for each language brain region separately, and report the results in the remaining columns of Table 2. We make the following observations: (1) Both TreeDepth and TopConstituents are responsible for the bump in brain alignment not just for the entire brain but also for several language ROIs. TopConstituents is important across all language ROIs, but most important in IFG which is related to syntax. This result aligns with previous work that has found the inferior frontal regions to be sensitive to syntax (Friederici et al., 2003; Friederici, 2012). (2) Unlike TopConstituents which has a strong effect on the alignment with all language ROI, TreeDepth has a strong effect only on alignment with the temporal (ATL and PTL) and frontal regions (IFG and MFG). (3) Although semantic properties are not shown to be responsible for the shape in brain alignment at the whole-brain level, semantic properties such as Tense, Subject Number and Object Number affect the alignment more specifically in PCC. This finding agrees with previous work which suggests the PCC supports word semantics (Binder et al., 2009). (4) The surface property

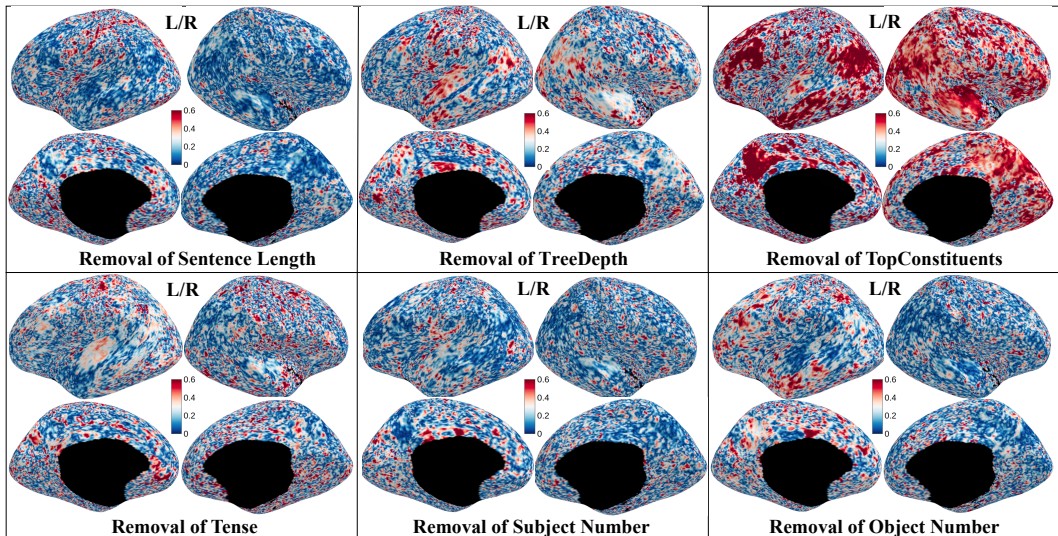

Figure 4: Voxel-wise correlations across layers between 1) differences in decoding task performance before and after removing each property and 2) differences in brain alignment of pretrained BERT before and after removing the corresponding linguistic property. Here L/R denotes the left and right hemispheres.

Sentence Length has a strong effect on the alignment with the IFG and MFG, but not as high as the syntactic properties in these language ROIs.

**Sub-ROI-Level Analysis** Each language brain region is not necessarily homogeneous in function across all voxels it contains. Therefore, an aggregate analysis across an entire language region may mask some nuanced effects. Thus, we further analyze several important language sub-regions that are thought to exemplify the variety of functionality across some of the broader language regions (Rolls et al., 2022): AG sub-regions (PFm, PGi, PGs), ATL sub-regions (STGa, STSda), PTL sub-regions (STSdp, A5, TPOJ1, PSL, STV, SFL), and IFG sub-regions (44, 45, IFJa, IFSp). We present the correlations for each of these sub-regions separately in Table 3. We make the following observations: (1) Although TopConstituents strongly affect the trend in the brain alignment across layers for the IFG, it strongly affects the trend for only three sub-regions (44, 45, and IFSp). It is possible that other factors affect the trend in alignment with IFJa across layers, as this IFG region is involved in many cognitive processes, including working memory for maintaining and updating ongoing information (Rainer et al., 1998; Zanto et al., 2011; Gazzaley & Nobre, 2012). (2) While TopConstituents was shown to have the strongest effect on the brain alignment with the whole IFG, the alignment with two of the IFG sub-regions (IFSp and IFJa) is more strongly affected by Tree Depth. (3) Subject Number has a medium to strong effect on the alignment with AG sub-region PFm. This implies that the semantic property Subject Number is important in at least one sub-region of language ROI. (4) Similarly, we find Object Number to be important for the trend of alignment across layers with most language sub-regions of ATL and PTL. Taken together, these results show that the semantic properties are most responsible for the trend in brain alignment across layers for many language sub-regions of ATL, PTL and AG, while syntactic properties have the highest effect on the alignment with the IFG.

**Qualitative Analysis** To present the correlations above at an even finer grain, we show them now at the voxel-wise level in Figure 4. We make the following qualitative observations: (1) These results confirm that Sentence Length and Subject Number are the least related to the pattern of the brain alignment across layers. (2) The effect of Tree Depth, Top Constituents, and Object Number is more localized to the canonical language regions in the left hemisphere and is more distributed in the right hemisphere. (3) The semantic property Object Number has a much larger effect on the alignment with the left hemisphere. Furthermore, Appendix Figure 12 shows the voxel-wise correlations across layers when removing all linguistic properties, instead of each individual one. We observe that removing all linguistic properties leads to very different region-level brain maps compared to removing individual

linguistic properties (Figure 4), and that the remaining correlation across layers after removing all linguistic properties is not substantial in the key language regions.

# 6 Discussion and Conclusion

We propose a direct approach for evaluating the joint processing of linguistic properties in brains and language models. We show that the removal of a range of linguistic properties from both language models (pretrained BERT and GPT2) leads to a significant decrease in brain alignment across all layers in the language model.

To understand the contribution of each linguistic property to the trend of brain alignment across layers, we leverage an additional analysis for the whole brain, language regions and language sub-regions: computing the correlations across layers between 1) differences in decoding task performance before and after removing each linguistic property and 2) differences in brain alignment of pretrained BERT before and after removing the corresponding linguistic property. For the whole brain, we find that Tree Depth and Top Constituents are the most responsible for the trend in brain alignment across BERT layers. Specifically, TopConstituents has the largest effect on the trend in brain alignment across BERT layers for all language regions, whereas TreeDepth has a strong effect on a smaller subset of regions that include temporal (ATL and PTL) and frontal language regions (IFG and MFG). Further, although other properties may not have a large impact at the whole brain level, they are important locally. We find that semantic properties, such as Tense, Subject Number, and Object Number affect the alignment with more semantic regions, such as PCC (Binder et al., 2009).

One limitation of our removal approach is that any information that is correlated with the removed linguistic property in our dataset will also be removed. This limitation can be alleviated by increasing the dataset size, which is becoming increasingly possible with new releases of publicly available datasets. It is also important to note that while we find that several linguistic properties affect the alignment between fMRI recordings and a pretrained language model, we also observed that there is substantial remaining brain alignment after removal of all linguistic properties. This suggests that our analysis has not accounted for all linguistic properties that are jointly processed. Future work can build on our approach to incorporate additional linguistic properties to fully characterize the joint processing of information between brains and language models. The current study was performed using experimental stimulus in English and language models trained on English text. While we expect our results to be generalizable at least to languages that are syntactically close to English, this should be explored by future work. Additionally, investigating larger language models beyond the two models used in our study, BERT and GPT-2, can contribute to an even more comprehensive understanding of the reasons for alignment between language models and human brain activity.

The insights gained from our work have implications for AI engineering, neuroscience and model interpretability–some in the short-term, others in the long-term. **AI engineering:** Our work most immediately fits in with the neuro-AI research direction that specifically investigates the relationship between representations in the brain and representations learned by powerful neural network models. This direction has gained recent traction, especially in the domain of language, thanks to advancements in language models (Schrimpf et al., 2021; Goldstein et al., 2022). Our work most immediately contributes to this line of research by understanding the reasons for the observed similarity in more depth. Specifically, our work can guide linguistic feature selection, facilitate improved transfer learning, and help in the development of cognitively plausible AI architectures. **Computational Modeling in Neuroscience:** Researchers have started viewing language models as useful *model organisms* for human language processing (Toneva, 2021) since they implement a language system in a way that may be very different from the human brain, but may nonetheless offer insights into the linguistic tasks and computational processes that are sufficient or insufficient to solve them (McCloskey, 1991; Baroni, 2020). Our work enables cognitive neuroscientists to have more control over using language models as model organisms of language processing. **Model Interpretability:** In the long-term, we hope that our approach can contribute to another line of work that uses brain signals to interpret the information contained by neural network models (Toneva & Wehbe, 2019; Aw & Toneva, 2023). We believe that the addition of linguistic features by our approach can further increase the model interpretability enabled by this line of work. Overall, we hope that our approach can be used to better understand the necessary and sufficient properties that lead to significant alignment between brain recordings and language models.

## Acknowledgments

This work was partially funded by the German Research Foundation (DFG) - DFG Research Unit FOR 5368. We thank Gabriele Merlin and Camila Kolling for providing valuable feedback on earlier versions of this work.

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
