# Appendix

## A Distribution of Class Labels Across Each Probing Task

Figure 5 reports the distribution of class labels across each linguistic property. For the probing tasks such as Sentence Length, Tree Depth, and Top Constituents, we balance the number of classes to overcome the imbalance problem. We balance the classes for these three probing tasks as follows: (i) Sentence Length: 3-classes ($\leq 5$, 5-8 and $\geq 9$), (ii) TreeDepth: 3-classes (5, 6-7 and $\geq 8$), and TopConstituents: 2-classes (1, $\geq 2$).

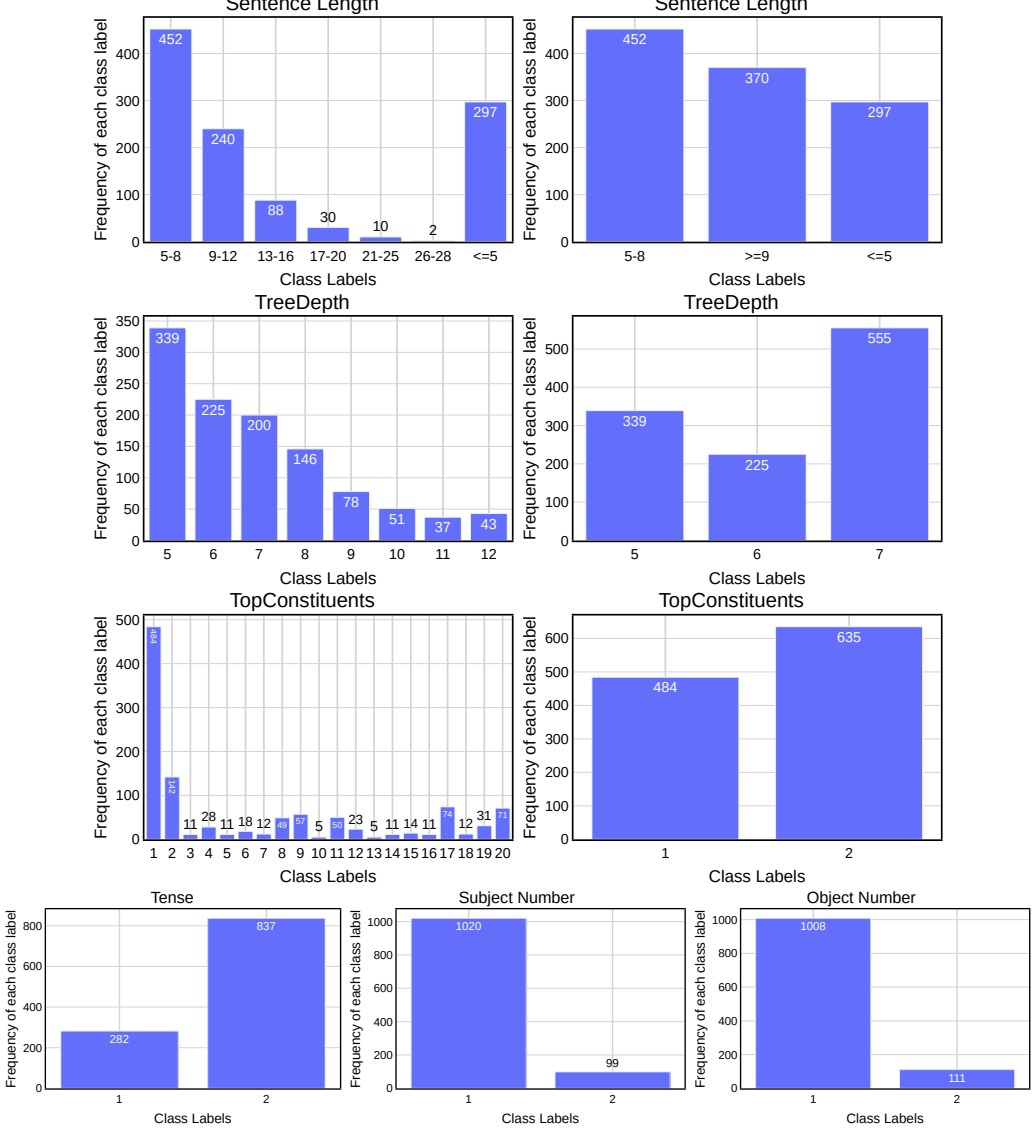

Figure 5: Distribution of class labels across each linguistic property. For the probing tasks such as Sentence Length, Tree Depth, and Top Constituents, we balance the number of classes to overcome the imbalance problem.

Figure 6 displays the common samples between class labels of pair of probing tasks. This reports whether the cells are balanced across class labels of different probing tasks.

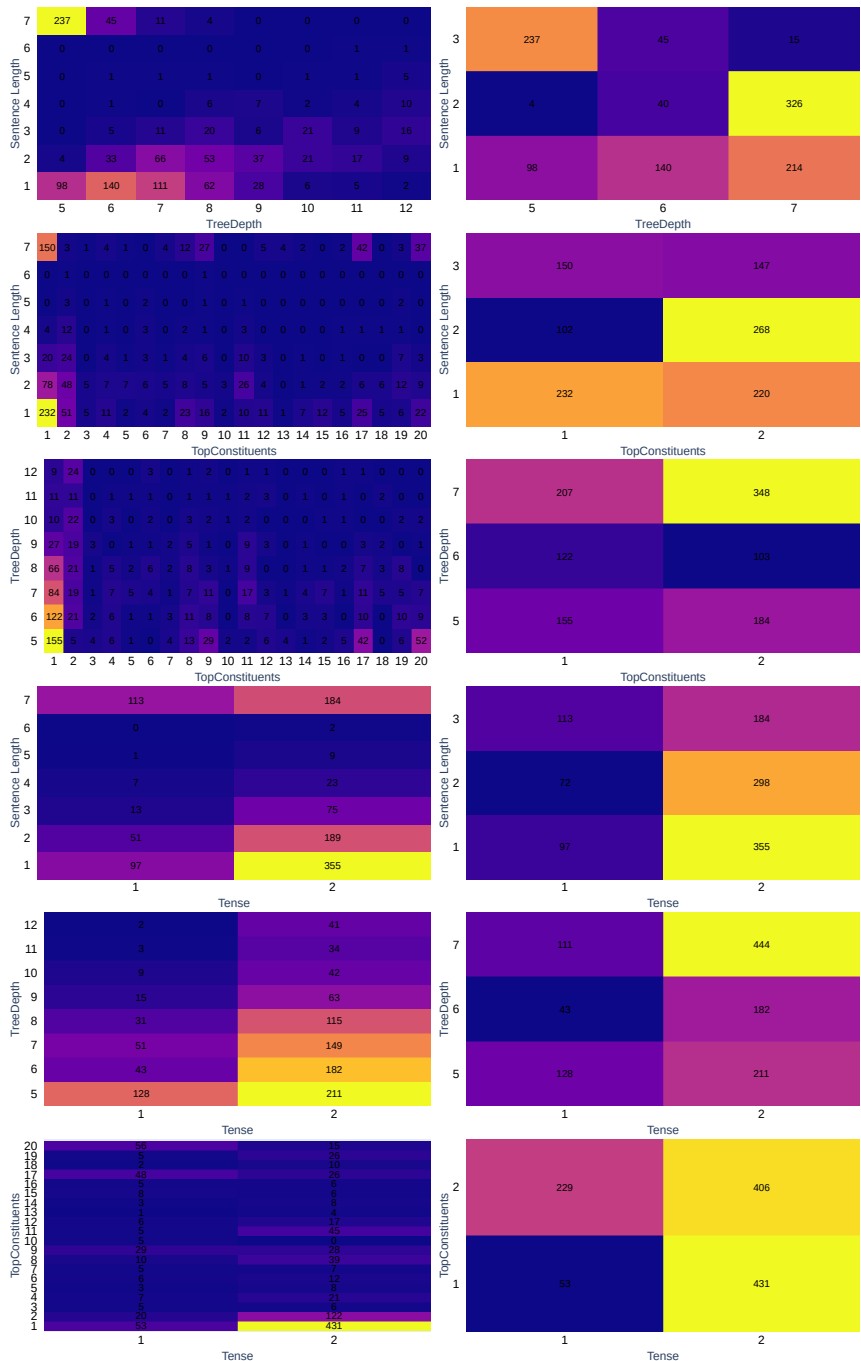

Figure 6: Number of common samples between pair of class labels across probing tasks.

# B    INLP Projection vs. Our Method

We also implemented the Iterative Null-Space Projection (INLP) method (Ravfogel et al., 2020) to verify whether our removal method performance is similar to previously proposed method. We found the results to be similar. Results using our method are in Table 4. Results using the INLP method are below.

Table 4: Word-Level: INLP Projection Method vs Our Removal Method: Probing task accuracy for each BERT layer after removal of each linguistic property using the $21^{st}$ year stimuli.

| Layers | Sentence Length | TreeDepth | TopConstituents | Tense | Subject Number | Object Number |
|---|---|---|---|---|---|---|
| 1 | 38.23 | 57.92 | 49.36 | 50.75 | 50.50 | 50.89 |
| 2 | 37.03 | 34.36 | 41.47 | 44.13 | 47.61 | 49.20 |
| 3 | 37.20 | 28.94 | 36.76 | 59.12 | 48.37 | 47.73 |
| 4 | 38.71 | 28.54 | 54.72 | 40.81 | 48.04 | 56.03 |
| 5 | 39.18 | 41.17 | 47.27 | 41.54 | 60.27 | 54.48 |
| 6 | 37.96 | 33.19 | 46.97 | 34.82 | 48.24 | 50.88 |
| 7 | 39.43 | 35.97 | 36.94 | 45.68 | 50.09 | 60.27 |
| 8 | 38.85 | 27.32 | 36.94 | 48.77 | 58.09 | 47.18 |
| 9 | 37.00 | 27.99 | 49.46 | 64.99 | 60.50 | 48.65 |
| 10 | 38.87 | 36.69 | 36.94 | 45.28 | 56.16 | 47.75 |
| 11 | 36.15 | 42.68 | 51.93 | 36.27 | 57.65 | 62.87 |
| 12 | 38.14 | 51.33 | 55.19 | 31.19 | 49.81 | 47.04 |
| Avg (INLP) | 38.06 | 37.17 | 45.32 | 45.27 | 52.94 | 51.91 |
| Avg (Ours) | 45.30 | 41.77 | 46.30 | 46.36 | 51.75 | 50.24 |
| Chance (Probability) | 42.76 | 50.39 | 53.63 | 66.31 | 83.24 | 80.78 |

## B.1 Probing Analysis While Removing Information about a Property and Testing Another

We run the probing analysis while removing information about a property and testing another, to see if only information specific about a task is being removed at a time. The detailed result of each probing task is presented Tables 5, 6, 7, 8, 9, and 10. We observe that removal of a task (property) does not affect the performance for other tasks except for TreeDepth and TopConstituents.

Table 5: Layer-wise Probing task performance for Sentence Length task. Note in tables below AR=After Removal and BR=before removal.

| Layers | BR Sentence Length | AR Sentence Length | AR TreeDepth | AR TopConstituents | AR Tense | AR SubjNum | AR ObjNum |
|---|---|---|---|---|---|---|---|
| 1 | **74.67** | 43.28 | 69.47 | 72.37 | 72.24 | 71.52 | 73.58 |
| 2 | 69.83 | 42.44 | 70.37 | 73.09 | 73.28 | 73.04 | 74.18 |
| 3 | 72.31 | 46.19 | 71.22 | 72.85 | 73.94 | 73.22 | 74.12 |
| 4 | 71.34 | 46.43 | 71.64 | 71.52 | 73.40 | 74.00 | 73.52 |
| 5 | 72.67 | 46.97 | 71.10 | 74.37 | 76.24 | 74.06 | 76.54 |
| 6 | 70.38 | 44.37 | 69.89 | 72.91 | 74.61 | 74.00 | 74.79 |
| 7 | 72.98 | 46.55 | 71.22 | 71.52 | 75.03 | 74.12 | 74.30 |
| 8 | 72.67 | 44.67 | 70.07 | 70.98 | 74.06 | 74.73 | 72.73 |
| 9 | 70.50 | 45.28 | 70.07 | 69.35 | 72.31 | 73.16 | 71.83 |
| 10 | 72.91 | 47.93 | 68.92 | 70.92 | 71.89 | 71.95 | 72.49 |
| 11 | 70.07 | 46.67 | 67.35 | 70.49 | 73.06 | 71.64 | 71.83 |
| 12 | 71.77 | 42.93 | 66.92 | 68.62 | 70.31 | 71.64 | 71.40 |
| Avg | 71.84 | 45.30 | 64.01 | 71.58 | 73.36 | 73.09 | 73.44 |

Table 6: Layer-wise Probing task performance for TreeDepth task. Note in tables below AR=After Removal and BR=before removal.

| Layers | BR TreeDepth | AR TreeDepth | AR SentenceLength | AR TopConstituents | AR Tense | AR SubjectNumber | AR ObjectNumber |
|---|---|---|---|---|---|---|---|
| 1 | 76.30 | 42.93 | 76.84 | 77.62 | 77.69 | 75.93 | 76.78 |
| 2 | 76.72 | 38.88 | 76.48 | 76.96 | 76.90 | 77.63 | 76.54 |
| 3 | 75.76 | 40.33 | 77.97 | 76.42 | 79.62 | 76.84 | 75.57 |
| 4 | 75.94 | 38.63 | 77.20 | 77.08 | 78.42 | 77.38 | 75.82 |
| 5 | 76.00 | 40.88 | 76.48 | 76.36 | 78.11 | 77.67 | 76.23 |
| 6 | **79.02** | 41.89 | 76.17 | 76.60 | 78.11 | 76.30 | 77.81 |
| 7 | 77.93 | 41.23 | 77.14 | 77.56 | 77.50 | 77.50 | 77.81 |
| 8 | 76.07 | 40.08 | 76.05 | 76.66 | 76.36 | 76.06 | 76.84 |
| 9 | 77.15 | 42.62 | 76.90 | 77.08 | 77.93 | 78.47 | 77.38 |
| 10 | 76.90 | 41.78 | 76.05 | 76.54 | 74.84 | 76.29 | 76.42 |
| 11 | 77.27 | 45.47 | 75.15 | 76.42 | 76.17 | 77.56 | 77.14 |
| 12 | 76.39 | 46.61 | 75.99 | 76.23 | 77.75 | 75.09 | 76.01 |
| Average | 76.79 | 41.77 | 76.56 | 76.70 | 77.75 | 75.09 | 76.70 |

## C Layer-wise Whole Brain Analysis before/after removal of linguistic properties.

In Figure 7, we report the layer-wise performance for pretrained BERT before and after the removal of each of the linguistic properties. Similar to previous work (Toneva & Wehbe, 2019), we observe that pretrained BERT has best brain alignment in the middle layers across all properties. We further observe that the alignment after removing the linguistic property is significantly worse mainly for middle to late layers. Note that the layers at which there is a significant difference are indicated with a red dot. This pattern holds across all of the linguistic properties that we tested. These results provide

Table 7: Layer-wise Probing task performance for TopConstituents task. Note in tables below AR=After Removal and BR=before removal.

| Layers | BR TopConstituents | AR TopConstituents | AR SentenceLength | AR TreeDepth | AR Tense | AR SubjectNumber | AR ObjectNumber |
|---|---|---|---|---|---|---|---|
| 1 | 77.15 | 47.28 | 78.17 | 78.11 | 76.72 | 77.39 | 77.81 |
| 2 | 78.60 | 42.75 | 77.69 | 78.54 | 78.41 | 77.81 | 77.93 |
| 3 | 77.81 | 48.85 | 76.36 | 77.15 | 77.15 | 77.69 | 77.75 |
| 4 | 78.36 | 48.00 | 77.69 | 77.03 | 77.81 | 78.23 | 77.69 |
| 5 | 78.60 | 45.28 | 77.87 | 78.42 | 78.35 | 77.93 | 78.36 |
| 6 | **80.23** | 43.47 | 78.71 | 79.20 | 77.21 | 78.54 | 78.65 |
| 7 | **80.23** | 46.43 | 78.65 | 78.66 | 77.39 | 79.02 | 79.38 |
| 8 | 78.90 | 46.86 | 77.63 | 78.36 | 78.41 | 79.08 | 79.44 |
| 9 | 79.87 | 44.55 | 77.76 | 78.30 | 78.71 | 79.87 | 78.89 |
| 10 | 78.17 | 47.76 | 77.81 | 77.87 | 78.41 | 78.96 | 78.42 |
| 11 | 77.69 | 45.77 | 78.42 | 78.84 | 77.03 | 78.00 | 78.00 |
| 12 | 78.29 | 48.67 | 76.36 | 78.78 | 75.21 | 78.30 | 78.36 |
| Average | 78.66 | 46.30 | 77.76 | 78.27 | 78.16 | 78.40 | 78.39 |

Table 8: Layer-wise Probing task performance for Tense task. Note in tables below AR=After Removal and BR=before removal.

| Layers | BR Tense | AR Tense | AR SentenceLength | AR TreeDepth | AR TopConstituents | AR SubjectNumber | AR ObjectNumber |
|---|---|---|---|---|---|---|---|
| 1 | 87.00 | 59.25 | 87.00 | 86.22 | 85.91 | 85.07 | 86.82 |
| 2 | 87.18 | 48.25 | 88.15 | 87.06 | 87.54 | 86.22 | 86.64 |
| 3 | 87.42 | 44.26 | 87.55 | 87.30 | 87.48 | 87.55 | 87.18 |
| 4 | 88.09 | 42.56 | 87.42 | 87.49 | 86.82 | 87.36 | 87.73 |
| 5 | 88.39 | 44.26 | 88.03 | 88.08 | 87.73 | 87.12 | 87.67 |
| 6 | 87.17 | 44.44 | 86.33 | 86.70 | 86.22 | 87.42 | 86.46 |
| 7 | 88.69 | 42.62 | 86.52 | 88.33 | 88.09 | 87.30 | 87.79 |
| 8 | 87.42 | 44.56 | 86.82 | 87.24 | 86.64 | 86.52 | 84.95 |
| 9 | 88.27 | 47.22 | 87.79 | 87.84 | 87.73 | 87.67 | 87.48 |
| 10 | **88.94** | 45.47 | 88.75 | 88.21 | 86.15 | 87.90 | 88.21 |
| 11 | 87.24 | 48.43 | 86.88 | 87.42 | 86.94 | 87.00 | 86.46 |
| 12 | 86.88 | 45.10 | 86.40 | 86.64 | 85.31 | 85.97 | 86.40 |
| Average | 87.72 | 46.36 | 87.30 | 87.38 | 86.88 | 86.93 | 86.98 |

Table 9: Layer-wise Probing task performance for Subject Number task. Note in tables below AR=After Removal and BR=before removal.

| Layers | BR SubjectNumber | AR SubjectNumber | AR SentenceLength | AR TreeDepth | AR TopConstituents | AR Tense | AR ObjectNumber |
|---|---|---|---|---|---|---|---|
| 1 | 87.00 | 59.25 | 87.57 | 87.76 | 87.88 | 87.56 | 71.61 |
| 2 | 92.32 | 55.50 | 86.40 | 86.57 | 86.45 | 86.15 | 69.82 |
| 3 | 93.04 | 48.55 | 86.84 | 86.94 | 86.77 | 86.91 | 70.17 |
| 4 | 93.50 | 50.12 | 86.56 | 86.52 | 86.68 | 86.97 | 70.84 |
| 5 | 94.05 | 49.88 | 87.78 | 87.87 | 87.59 | 87.70 | 70.07 |
| 6 | 94.98 | 55.08 | 87.59 | 87.77 | 87.76 | 87.88 | 70.96 |
| 7 | 95.88 | 50.24 | 87.84 | 87.94 | 87.56 | 87.76 | 70.66 |
| 8 | 96.10 | 50.24 | 87.35 | 87.65 | 87.04 | 86.85 | 71.13 |
| 9 | 96.38 | 52.78 | 87.52 | 87.55 | 87.32 | 87.09 | 70.50 |
| 10 | 96.06 | 53.68 | 87.78 | 8.07 | 87.80 | 87.31 | 70.91 |
| 11 | **96.94** | 53.44 | 87.29 | 87.49 | 87.47 | 86.97 | 70.81 |
| 12 | 94.03 | 51.45 | 87.44 | 87.55 | 87.64 | 87.23 | 70.84 |
| Average | 94.19 | 51.75 | 87.33 | 87.46 | 87.33 | 87.20 | 70.69 |

Table 10: Layer-wise Probing task performance for Object Number task. Note in tables below AR=After Removal and BR=before removal.

| Layers | BR ObjectNumber | AR ObjectNumber | AR SentenceLength | AR TreeDepth | AR TopConstituents | AR Tense | AR SubjectNumber |
|---|---|---|---|---|---|---|---|
| 1 | 93.28 | 47.31 | 86.63 | 86.68 | 86.77 | 86.83 | 71.45 |
| 2 | 93.47 | 54.59 | 86.55 | 86.61 | 86.63 | 86.55 | 70.20 |
| 3 | 93.80 | 49.76 | 86.88 | 86.84 | 86.85 | 86.62 | 70.15 |
| 4 | 94.90 | 50.06 | 85.81 | 85.87 | 85.87 | 85.79 | 71.14 |
| 5 | 93.59 | 51.45 | 86.17 | 86.42 | 86.48 | 86.48 | 70.05 |
| 6 | 94.50 | 54.17 | 86.27 | 86.45 | 86.19 | 86.74 | 70.61 |
| 7 | 94.62 | 47.58 | 86.38 | 86.42 | 86.33 | 86.39 | 71.00 |
| 8 | **95.10** | 50.18 | 85.86 | 85.79 | 85.79 | 85.57 | 70.76 |
| 9 | 94.56 | 49.27 | 85.63 | 85.61 | 85.55 | 85.46 | 70.20 |
| 10 | 94.50 | 50.30 | 86.21 | 86.16 | 86.21 | 86.16 | 70.73 |
| 11 | 94.92 | 49.52 | 84.94 | 84.95 | 84.90 | 85.01 | 70.33 |
| 12 | 93.95 | 48.73 | 85.49 | 85.33 | 85.65 | 85.30 | 70.49 |
| Average | 94.27 | 50.24 | 86.07 | 86.11 | 86.10 | 86.09 | 70.59 |

direct evidence that these linguistic properties in fact significantly Figure 7 affect the alignment between fMRI recordings and pretrained BERT.

# D   Layer-wise Language ROIs Analysis before/after removal of linguistic properties.

Here, we examine the effect on the alignment specifically in a set of regions of interest (ROI) that are thought to underlie language comprehension (Fedorenko et al., 2010; Fedorenko & Thompson-Schill, 2014) and word semantics (Binder et al., 2009). We find that across language regions, the alignment is significantly decreased by the removal of the linguistic properties across all layers, as shown in Figure 8.

# E   Layer-wise Language sub-ROIs Analysis before/after removal of linguistic properties.

We further investigate the language sub-regions, such as 44, 45, IFJa, IFSp, STGa, STSdp, STSda, A5, PSL, and STV and find that they also align significantly worse after the removal of the linguistic properties from pretrained BERT (see Figure 9).

Each language brain region is not necessarily homogeneous in function across all voxels it contains. Therefore, an aggregate analysis across an entire language region may mask some nuanced effects. We thus further analyze several important language sub-regions that are thought to exemplify the variety of functionality across the broader language regions (Rolls et al., 2022): STGa, STSda, 44, 45, 55b, STV, SFL, PFm, PGi, PGs, IFJa and IFSp in the main paper. Here we extend this analysis to more sub-regions: PSL, STV, SFL, PFm, PGs, PGi. We demonstrate the correlations for each language brain sub-region separately in Table 3 in the main paper.

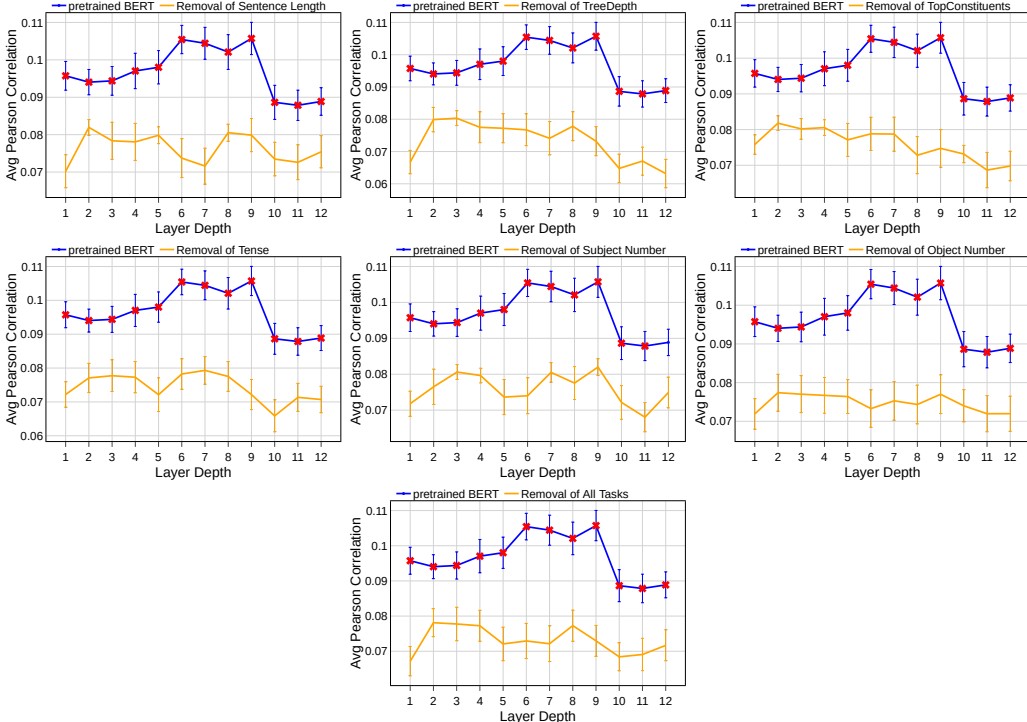

Figure 7: Balanced Classes: Model trained on pretrained BERT features and removal of different linguistic properties. Each plot compares the layer-wise performance of pretrained BERT and removal of each probing task. Bottom plot displays the pretrained BERT vs. removal of all tasks. A red ∗ at a particular layer indicates that the alignment from the pretrained model is significantly reduced by the removal of this linguistic property at this particular layer.

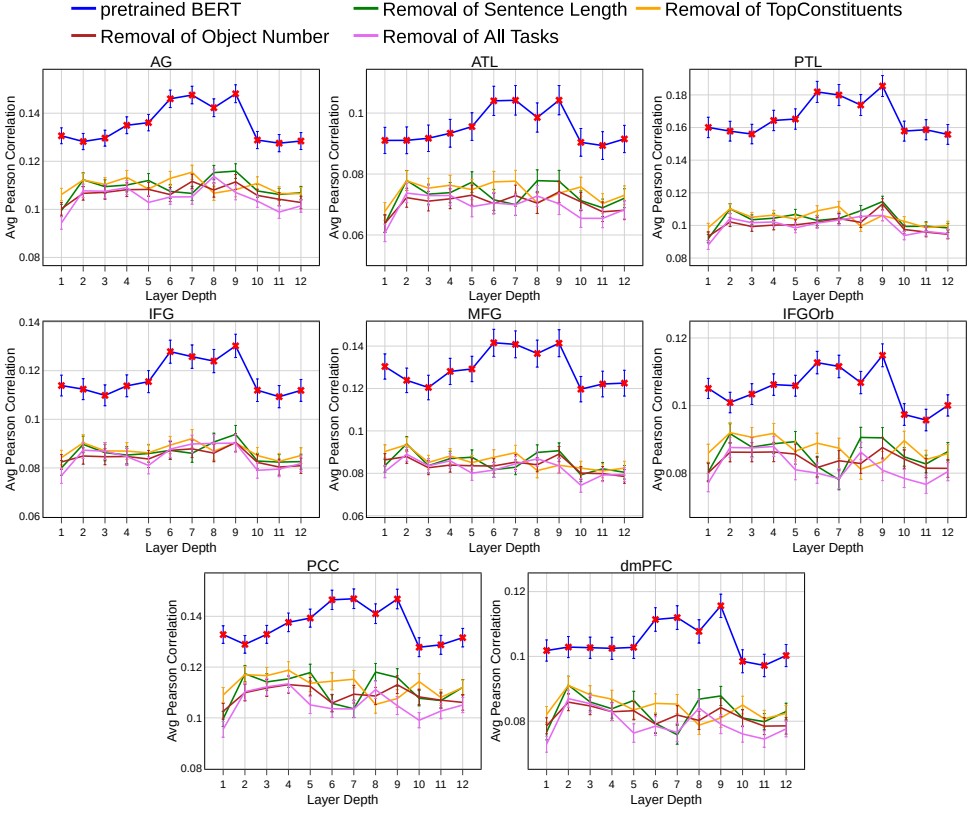

Figure 8: Avg Pearson correlations for language regions AG, ATL, PTL, IFG, MFG, IFGOrb, PCC and dmPFC. Model trained on pretrained BERT features and removal of different linguistic properties (shown only for important properties for clarity). Each plot compares the layer-wise performance of pretrained BERT and removal of each probing task.

## F    Probing Results: GPT2

Like the probing experiments with BERT in the main paper, we also perform experiments with GPT2. We find the results to be similar to BERT, i.e., a rich hierarchy of linguistic signals: initial to middle layers encode surface information, middle layers encode syntax, middle to top layers encode semantics. Table 11 reports the result for each probing task, before and after removal of the linguistic property from pretrained GPT2. We verify that the removal of each linguistic property from GPT2 leads to reduced task performance across all layers, as expected. We also report the layer-wise performance for pretrained GPT2 before and after the removal of one representative linguistic property (TopConstituents) in Fig. 11. We observe that the brain alignment is reduced significantly across all layers after the removal of the linguistic property (indicated with red cross in the figure).

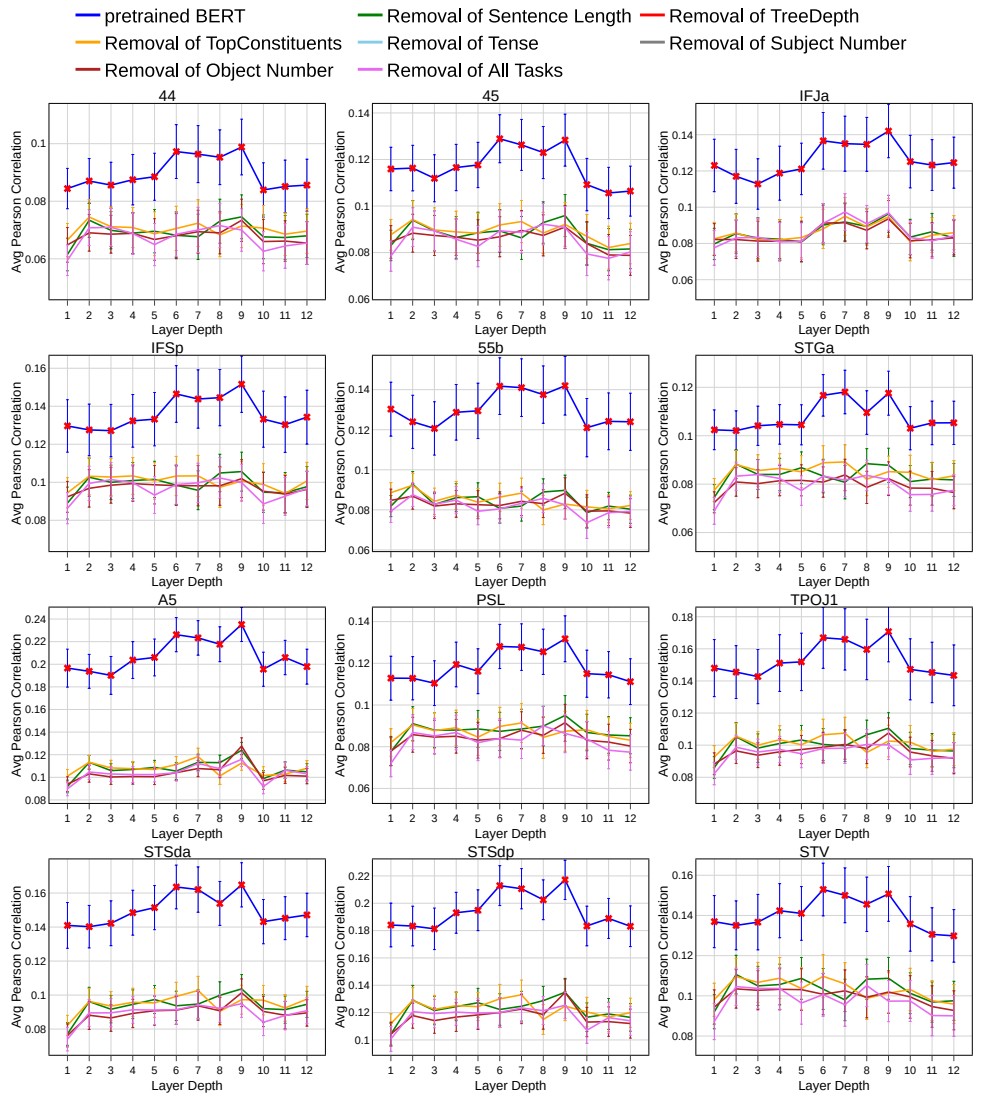

Figure 9: We extend analyses in Fig. 7 to some language sub-regions 45, IFJa, IFSp, 55b, STGa, A5, PSL, TPOJ1, STSda, STSdp and STV. Please refer to caption of Fig. 7 for detailed understanding of each plot.

Table 11: Word-Level Probing task performance for each GPT2 layer before and after removal of each linguistic property using the $21^{st}$ year stimuli.

| Layers | Sentence Length 3-classes (Surface) | | TreeDepth 3-classes (Syntactic) | | TopConstituents 2-classes (Syntactic) | | Tense 2-classes (Semantic) | | Subject Number 2-classes (Semantic) | | Object Number 2-classes (Semantic) | |
|---|---|---|---|---|---|---|---|---|---|---|---|---|
| | before | after | before | after | before | after | before | after | before | after | before | after |
| 1 | **54.11** | 30.48 | 65.78 | 32.78 | 60.88 | 45.56 | 76.59 | 49.78 | 83.57 | 49.90 | 87.17 | 50.89 |
| 2 | 54.00 | 32.50 | 66.62 | 33.65 | 60.52 | 44.55 | 77.57 | 50.79 | 87.87 | 50.91 | 87.60 | 50.88 |
| 3 | 53.02 | 32.51 | 66.26 | 33.67 | 60.46 | 49.63 | 77.97 | 50.78 | 87.94 | 50.91 | 87.97 | 50.91 |
| 4 | 52.90 | 33.54 | 67.38 | 34.64 | 60.94 | 49.63 | 77.98 | 50.78 | 88.84 | 50.51 | 88.10 | 52.12 |
| 5 | 52.94 | 32.53 | 67.78 | 33.46 | 61.43 | 42.34 | 77.85 | 50.39 | 89.03 | 51.05 | 88.42 | 51.87 |
| 6 | 52.47 | 33.55 | 68.14 | 33.23 | **62.21** | 48.62 | 78.17 | 50.79 | 89.50 | 50.90 | 88.81 | 50.31 |
| 7 | 52.24 | 33.53 | **68.68** | 34.55 | 61.43 | 48.38 | 78.02 | 51.68 | 89.89 | 50.95 | 88.98 | 50.88 |
| 8 | 52.00 | 33.52 | 68.08 | 31.22 | 61.43 | 49.39 | 78.21 | 51.81 | 89.81 | 51.81 | 89.10 | 52.12 |
| 9 | 53.03 | 32.53 | 67.47 | 33.01 | 61.19 | 49.62 | **78.22** | 51.17 | 90.23 | 50.91 | **89.14** | 52.75 |
| 10 | 52.47 | 32.52 | 67.53 | 33.67 | 61.37 | 48.61 | 78.20 | 49.57 | 90.17 | 49.39 | 89.09 | 51.73 |
| 11 | 52.53 | 32.50 | 67.47 | 34.64 | 61.19 | 48.63 | 77.96 | 50.78 | **90.56** | 51.63 | 89.08 | 52.53 |
| 12 | 52.83 | 33.53 | 66.44 | 34.27 | 61.06 | 48.61 | 77.62 | 49.78 | 90.10 | 51.07 | 89.06 | 54.34 |

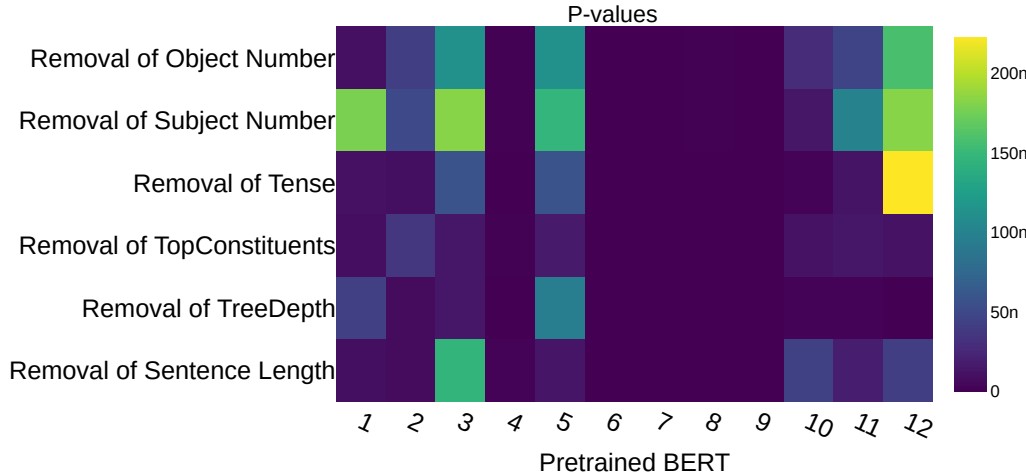

Figure 10: For whole brain, paired t-test was performed to identify the layers and removal of linguistic properties where the pretrained BERT model has greater brain alignment than the removal of each linguistic property, with statistical signifi- cance. p-values obtained from paired t-test, after false discovery rate (FDR) correction using the Benjamini–Hochberg (BH) pro- cedure.

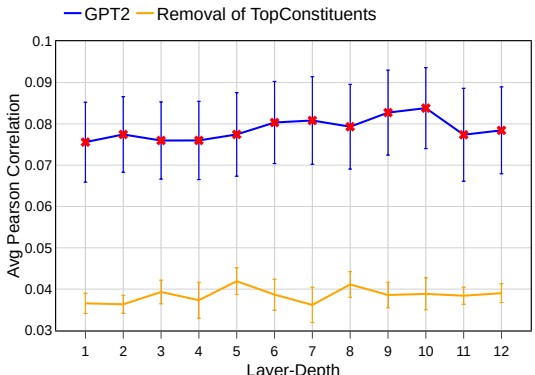

Figure 11: Comparison of layer-wise performance of pretrained GPT2 and removal of one linguistic property–TopConstituents.

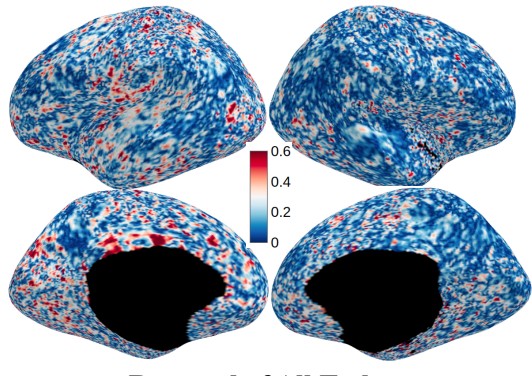

**Removal of All Tasks**

Figure 12: Voxel-wise correlations across layers between brain alignment of pretrained BERT before and after removing all linguistic properties.