# OpenReview forum: "Joint processing of linguistic properties in brains and language models"
_NeurIPS.cc/2023/Conference — NeurIPS 2023 poster_

### Official Review · Reviewer_p9mD · 2023-06-29

**Soundness:** 3 good
**Presentation:** 3 good
**Contribution:** 3 good
**Rating:** 7
**Confidence:** 4

**Summary:**

While several papers have recently shown that large language model embeddings are aligned with / predictive of fMRI reading data, this paper pushes these analyses forward by asking: what properties of the representations are responsible for this alignment?  In particular, they compare LLM/fMRI alignment on reading data using BERT, but then compare this to BERT's representations _when certain linguistic properties have been removed_ (using both a novel and an existing removal method).  The results show a significant decrease in alignment (crucially, not found when removing "random" information or when using random vectors) across all linguistic properties, suggesting that it really is BERT's encoding of linguistic information that's responsible for the alignment with fMRI reading data.  This represents a finer-grained level of analysis than existing alignment works and should be of interest to many in the field.

**Strengths:**

- Analyzes LLM representations' ability to predict fMRI data before and after "removing" linguistic information from said representations.
- Conducts analyses at both macro (whole-model, and whole-brain) as well as micro (layer-wise, ROI and sub-ROI) levels.

**Weaknesses:**

- The Pearson correlations in Figure 3 are quite low, on average less than 0.1.  What do the authors make of this?  The ROI-level analyses seem relevant, but are not done with the same metric, so direct comparison is hard.
- Linguistic annotations for the text in the reading data come from an off-the-shelf NLP tool (stanza), so these are "silver" rather than "gold" annotations.
- More baselines would have been useful.

**Questions:**

- Table 1: what are baselines for each task?  What would a most-frequent-class baseline score on each?  This would be a helpful comparison for the post-removal numbers.
- Word length probing: I'm a bit confused here.  Probing for word length should be hard for BERT since it does not have character-level information but instead uses sub-word tokenization.  I noticed in the Appendix that the task is referred to as "Sentence Length".  Is the task length-of-individual-word (that's what "Word Length" sounds like to me) or is it number-of-words-in-sentence ("Sentence Length")?  Consistency with naming, and clarity on the exact task, would be welcome here.
- Why the choice of BERT as the model for comparison?  In particular, I am curious about the choice of a "bidirectional" masked language model as opposed to a "left-to-right" language model like the GPT series.  Some would argue that the latter more closely resemble the reading process, and so might also better predict data from reading.


### Typographic comments:

- Line 125 needs a closing period.
- Lines 185 and 186: "z-scored" -> "$z$-scored".  Same for "t-test" and "p-value" throughout the paper.

**Limitations:**

- Use of stanza corenlp for linguistic annotations; these are "silver", not "gold".
- Only looked at one model, with relatively weak overall correlations with brain activity.

---

> ### Author Rebuttal · Authors · 2023-08-09
>
> *We thank the reviewer for their strong positive, insightful and valuable comments and suggestions which are crucial for further strengthening our manuscript.*
>
> **1. The Pearson correlations in Figure 3 are quite low, on average less than 0.1. What do the authors make of this? The ROI-level analyses seem relevant, but are not done with the same metric, so direct comparison is hard.**
>
> * The Pearson correlation results reported in our work are in similar ranges as reported in previous studies.
> * The correlations are low but significant.
> * We performed a two-tailed hypothesis test for all the tasks and did the Benjamni-Hochberg False Discovery Rate (FDR) correction for multiple comparisons.
> * One primary reason for the low correlation is partially due to noise in the brain recordings. If one estimates a noise ceiling, we see that these numbers correspond to about 70% of the explainable variance by a model representation.
>
> **2. Linguistic annotations for the text in the reading data come from an off-the-shelf NLP tool (stanza), so these are "silver" rather than "gold" annotations.**
>
> * We agree that the annotations are silver but since current NLP tools show high accuracies, we can assume them to be fairly accurate as done in many other related papers [1], [2].
>
> [1] Ganesh Jawahar, Benoît Sagot, Djamé Seddah. What does BERT learn about structure of language, ACL-2019
>
> [2[ Hosein Mohebbi, Ali Modarressi, Mohammad Taher Pilehvar. Exploring the Role of BERT Token Representations to Explain Sentence Probing Results, EMNLP-2021
>
>
> **3. Table 1: what are baselines for each task? What would a most-frequent-class baseline score on each? This would be a helpful comparison for the post-removal numbers**
>
> * Unfortunately, there is no previous work to compare with. We reported the chance performance of each probing task in Table 4 in Appendix.
>
> * It shows that after removal of a linguistic property our decoding task accuracy is close to chance performance showing that our property removal method works as expected. Also, Figure 5 displays the distribution of class labels in each of the probing tasks.
>
> **4. Word length vs Sent Length**
>
> * It is sentence length, thanks for pointing out the discrepancy. We will correct this.
>
> **5. Why the choice of BERT as the model for comparison over GPT2**
>
> * Please check “Common responses”, and Table 1 and Fig 1 in rebuttal PDF.
> * The primary rationale behind our choice to incorporate the BERT model comes from the extensive body of research that aims to delve into the extent to which diverse English language structures are embedded within Transformer-based encoders, a field often referred to as "BERTology" [3] (as highlighted by Jawahar et al., 2019, and Mohebbi et al., 2021). We will clarify this.
>
> [3] Rogers A, Kovaleva O, Rumshisky A. A primer in BERTology: What we know about how BERT works. Transactions of the Association for Computational Linguistics. 2021 Jan 1;8:842-66.
>
> **6. Typos**
>
> * We will address these typos.

---

> > ### Comment · Reviewer_p9mD · 2023-08-18
> >
> > I thank the authors for their detailed reply here and to everyone.  I think the inclusion of GPT2 results is very welcome, as is the clarification on the baseline (I would like to see that in the body, not an appendix, in the full version).  For now, I am happy to keep my score at 7 for a solid accept, but also would actively advocate for acceptance if there is disagreement with others.

---

> > > ### Author Response · Authors · 2023-08-19
> > >
> > > We appreciate the reviewer's feedback and are confident that it has enhanced the paper's quality.

---

### Official Review · Reviewer_9Zaj · 2023-07-03

**Soundness:** 2 fair
**Presentation:** 3 good
**Contribution:** 1 poor
**Rating:** 2
**Confidence:** 4

**Summary:**

This paper aims to investigate the correspondence between HIPSs and models based on Neural Networks such as the transformers. The key innovation of this paper is to study this correspondence by elimiating specicific linguistic properties form BERT and to observe how this intervation affects the alignment with fMRI brain recording.

**Strengths:**


- This is really an innovative standpoint

**Weaknesses:**

- There is a major innovation of this paper that is the cornerstone of the theory. Yet, this is not listed as a major result: the removal of linguistic properties from Transformers. Indeed, I was expecting citations to other papers as this is not listed in the main results of the paper. Yet, there are not citations and it seems to be proposed in this paper.


- Since the removal of linguistic properties from Transformers is a major cornerstone, all the theory could crumble if it is not investigated in a deep and correct way.

- Experiments to demonstrate that the removal of linguistic properties from Transformers are inconclusive. Indeed, Table 1 is not sufficient. It only says that after the adpatation BERT weights are so mess up that the classification task cannot be performed. There should be a comparative part or a sort of proof that the model is not compromised in the other parts. For example, there is a missing column: BERT with random parameters. Indeed, if the "after" models behave as BERT with random inizialization, results are not relevant.



**Questions:**

see weaknesses

---

> ### Author Rebuttal · Authors · 2023-08-09
>
> We thank the reviewer for their valuable comments and suggestions which are crucial for further strengthening our manuscript.
>
> **1. There is a major innovation of this paper that is the cornerstone of the theory. Yet, this is not listed as a major result: the removal of linguistic properties from Transformers. Indeed, I was expecting citations to other papers as this is not listed in the main results of the paper. Yet, there are not citations and it seems to be proposed in this paper.**
>
> * For the removal of linguistic property information from the model representations, we use two previously published approaches: residual method [1] and INLP method [2].
>
> * These approaches have been peer reviewed in good venues (Nature Computational Science and AAAI). We will clarify this.
> * We also conducted validation experiments to ensure that these removal techniques work in our setting (Please see the Appendix Tables 5-10).
>
> [1] Mariya Toneva, Tom M. Mitchell, Leila Wehbe. Combining computational controls with natural text reveals aspects of meaning composition, Nature Computational Science 2022.
>
> [2] Zhang X, Wang S, Lin N, Zhang J, Zong C. Probing word syntactic representations in the brain by a feature elimination method. In AAAI 2022..
>
>
> **2. Since the removal of linguistic properties from Transformers is a major cornerstone, all the theory could crumble if it is not investigated in a deep and correct way.**
>
> * To investigate whether a linguistic property was successfully removed from the pretrained representations, we further test whether we can decode the task labels from the residuals for each probing task of the 21st year stimuli.
> * To ensure that removal of a specific linguistic property works well, we show results in Table 1 (of the paper) by decoding from BERT before and after removal, decoding of linguistic property B when removing linguistic property A.
> * We observed that most properties are not substantially affected by removing other task properties.
> * Further, we observe that the brain alignment of a random vector is significantly lower than the residual brain alignment after removal of each linguistic property.
>
> **3. Proof that the model is not compromised in the other parts. For example, there is a missing column: BERT with random parameters.**
>
> * Fig 3 (in the paper) shows random embeddings are way worse compared to any of the “after” models. Table 1 (in the paper) and Appendix Tables 5-10 show that removing property A does not destroy the decoding performance from BERT on property B so therefore the embedding still contains important information.
> * Based on the reviewers’ suggestion, we now perform brain encoding experiments with random weights of BERT. Please check the *Fig 3 in rebuttal PDF* at “Common responses”.
> * Fig 3 (*in the rebuttal PDF*) shows brain predictivity performance using BERT with random weights is significantly worse compared to any of the “after” removal of linguistic property from pretrained BERT model. This shows that residual representations (i.e. removing a linguistic property from pretrained BERT) are informative and meaningful compared to random weights of BERT.

---

### Official Review · Reviewer_YHJW · 2023-07-05

**Soundness:** 3 good
**Presentation:** 3 good
**Contribution:** 2 fair
**Rating:** 5
**Confidence:** 4

**Summary:**

The paper investigates the relationship between specific linguistic properties and brain alignment across layers of a language model. The authors provide evidence for the importance of a linguistic property to the brain alignment. The focus of the study is to understand the degree to which various linguistic properties impact the performance of language models in predicting brain recordings.


**Strengths:**

**Originality:** The paper offers an interesting perspective on the link between language properties and brain alignment in language models. It builds on existing work but adds a new dimension by directly showing the importance of specific linguistic properties.

**Quality:** The paper uses a large, public fMRI dataset and word features from the BERT model to get word-level representations. The results are interesting.

**Clarity:** The paper is well-structured and easy to understand. It explains the methods and results clearly, making it accessible to a wide range of readers.

**Significance:** The approach paves a path towards a way to understand how linguistic properties and brain alignment relate. This could help improve language models and give new insights in neuroscience and language processing.


**Weaknesses:**

**Choice of Language Model:** The authors have chosen to focus on BERT, an encoder-only language model, for their study. Given that previous research has shown a deeper connection between autoregressive language models and the brain, the choice of BERT might limit the generalizability of the findings. The authors could consider including autoregressive language models in their analysis to provide a more comprehensive understanding of the relationship between linguistic properties and brain alignment.

**Correlation vs. Causation:** The authors aim to identify the effect of linguistic properties on brain alignment, but they rely on correlational metrics, which do not provide evidence of causation. To address this, the authors could consider methods from the causality+NLP literature to produce causal model explanations. This would strengthen their findings and provide more definitive evidence of the impact of linguistic properties on brain alignment. As it currently stands, it’s unclear which conclusions can be drawn from this study.

**Projection Method:** The authors use a projection method to identify the effect of linguistic information, which has been shown to be problematic and potentially misleading (Ravfogel et al., 2022). The authors could consider alternative methods for identifying the effect of linguistic information to ensure the accuracy and reliability of their findings. Some potential approaches include causal abstractions, which aim to provide a low-dimensional causal representation of LLMs.



**Questions:**

1. **Choice of Language Model:** Could you elaborate on why you chose to focus only on BERT, an encoder-only language model, for your study? Given that previous research has shown a deeper connection between autoregressive language models and the brain, have you considered including autoregressive language models in your analysis? How do you think the inclusion of such models might affect your findings?

2. **Correlation vs. Causation:** Your study aims to identify the effect of linguistic properties on brain alignment, but it relies on correlational metrics. Could you consider methods from the Causality+NLP literature to produce causal model explanations? How do you think a causal analysis might change your findings or conclusions?

3. **Projection Method:** You use a projection method to identify the effect of linguistic information, which has been shown to be problematic and potentially misleading (Ravfogel et al., 2022). Could you consider alternative methods for identifying the effect of linguistic information? How do you think a different method might affect your findings?


**Limitations:**

The authors do recognize some important limitations:

1. **Removal Approach:** They note that their method might unintentionally remove information correlated with the linguistic property they're studying. They suggest using larger datasets to mitigate this.

2. **Unaccounted Linguistic Properties:** They admit that there's still significant brain alignment even after removing all studied linguistic properties, indicating that there might be other linguistic properties at play that they haven't considered.

3. **Single Model Use:** They acknowledge that they've only tested their findings on one language model (BERT) and suggest that future work should test other models.

---

> ### Author Rebuttal · Authors · 2023-08-09
>
> *We thank the reviewer for their positive, insightful and valuable comments and suggestions which are crucial for further strengthening our manuscript.*
>
> **1. Choice of Language Model: BERT vs autoregressive models (GPT2)**
>
> * Please check “Common responses”, and Table 1 and Fig 1 in the rebuttal PDF.
> * The primary rationale behind our choice to incorporate the BERT model comes from the extensive body of research that aims to delve into the extent to which diverse English language structures are embedded within Transformer-based encoders, a field often referred to as "BERTology" [1] (as highlighted by Jawahar et al., 2019, and Mohebbi et al., 2021). We will clarify this.
> * We also additionally reported six probing tasks results for GPT2 before and after removal in the rebuttal PDF.
>
> [1] Rogers A, Kovaleva O, Rumshisky A. A primer in BERTology: What we know about how BERT works. Transactions of the Association for Computational Linguistics. 2021 Jan 1;8:842-66.
>
> **2. Correlation vs. Causation**
>
> * Thanks for the suggestion. Exploring the distinction between correlation and causation is an interesting avenue for future research work, and we will add it to the discussion.
> * We still believe that our work is a substantial advancement in this area as it allows us to perturb linguistic representations from models and observe the outcome of this perturbation on brain alignment.
>
> **3. Use of Projection Method to identify the effect of linguistic information**
>
> * We have used two methods (residual method and INLP method) to remove information related to linguistic properties from the model representations, and we obtained similar results with both techniques.
> * We use a linear projection method because the brain alignment function is also a linear function.
> * We understand the concerns raised in previous work that more powerful decoding functions may be able to reconstruct some information about the linguistic property, but what we care about is the linear information, since that is what is used to predict the brain response.
>
> * Additionally, our work is most closely related to that of [2], who employ a similar residual approach to study the supra-word meaning of language by removing the contribution of individual words to brain alignment.
>
> [2] Mariya Toneva, Tom M. Mitchell, Leila Wehbe. Combining computational controls with natural text reveals aspects of meaning composition, Nature Computational Science 2022.

---

> > ### Comment · Reviewer_YHJW · 2023-08-15
> >
> > Thanks for the rebuttal, I appreciate the time and effort, especially in adding the auto-regressive model experiments. While this paper has some flaws, I also think that is insightful and interesting. I increased my score.

---

> > > ### Author Response · Authors · 2023-08-16
> > > **Thank you for the reviewer's response**
> > >
> > > We appreciate the reviewer's feedback and are confident that it has enhanced the paper's quality.

---

### Official Review · Reviewer_wwF4 · 2023-07-07

**Soundness:** 4 excellent
**Presentation:** 4 excellent
**Contribution:** 3 good
**Rating:** 7
**Confidence:** 4

**Summary:**

The authors provide an analysis of correlations between BERT representations and fMRI data when abstracting away different linguistic properties from the BERT representations. They provide an in-depth analysis of the ways in which elimination of word length, tree depth, top constituents, tense, subject number and object number have on the overall and layer-wise alignment of BERT with fMRI data from 18 people. The authors find that removing any of these linguistic properties from the BERT representations reduces fMRI data alignment across all layers in the BERT model. Furthermore, syntactic properties are most relevant for the BERT-brain alignment globally, but semantic properties are highly locally aligned with brain regions generally associated with semantic processing.

**Strengths:**

The overall question and method is relevant to the community. This paper contributes to the cognitive debate on the overall alignment that today's language models have with people's cognitive systems, here concretely their linguistic neural processing of stories. By abstracting away a number of linguistic properties from the BERT representations in a controlled way, the results speak for the type of information encoded in BERT that lead to the reasonably high alignment with fMRI data. The work therefore contributes to a better understanding of BERT representations and the usefulness of investigating them as cognitive systems.

The authors integrate valuable controls into their study, such as random baselines and correlational analyses of tasks that nicely contextualize the findings.

The paper is very well-written.

**Weaknesses:**

To me, section 5.3 (Decoding task performance vs. brain alignment) has a number of interesting observations but I'm missing the overall motivation for this section. The authors write "While the previous analyses revealed the importance of linguistic properties for brain alignment at individual layers, we would also like to understand the contribution of each linguistic property to the trend of brain alignment across layers." (line 265ff). There seems to be a close connection between this question and Figure 3a from the previous section and it would be helpful to emphasize what the previous section lacks. To be clear, I think section 5.3 adds a lot, especially when it comes to the alignment of local instead of global brain regions but I currently can't find a motivation for whom this analysis is most relevant -- does this tell us something about BERT or are there any potential insights for neuroscience? The semantic alignment results seem to be backed up by prior findings in neuroscience. Is the same true for the rest, or if not, what does that mean?

BERT's layer-wise analysis of the correlations with fMRI data are central in the overall framing and centered in the main paper. I think that's rightfully so but I would like to encourage the authors to consider a strategy of substantiating claims that "BERT embeds a rich hierarchy [with] surface information at the initial and middle layers, syntactic information in the middle to top layers [...]" (lines 218ff). From inspecting the corresponding results table (Table 1), surface information seem maybe slightly more represented in the first and least in the second layer, but all other layers seem to be fairly evenly distributed. For the syntactic information, it seems uniformly high, and only subject number and object number seem to have a fairly monotonously increasing and decreasing structure that might suggest that it can't simply be due to random variance. Is there any way to estimate the variance of these terms or find a definition for when something is robustly represented more in some than other layers?

**Questions:**

I find Figure 3 very intriguing. Removing any linguistic property results in a correlation drop from 0.1 to about 0.75. The authors also previously added a correlational analysis indicating that most linguistic properties aren't correlated with each other. Consequently, it would make sense to me that when one removes all linguistic properties, the correlation needs to drop further since different sources of variance have been removed which don't correlate with each other but clearly correlate with the fMRI data. However, even this correlation is at about 0.75. Do you have an explanation why this might be?

Picking up on a notion in the "Weaknesses" section: Why doesn't figure 3 already provide us with an answer to the central motivation of section 5.3 ("we would also like to understand the contribution of each linguistic property to the trend of brain alignment across layers.")? Or did you mean to motivate section 5.3 with "across brain regions"?

**Limitations:**

Yes

---

> ### Author Rebuttal · Authors · 2023-08-09
>
> *We thank the reviewer for their strong positive, insightful and valuable comments and suggestions which are crucial for further strengthening our manuscript.*
>
> **1. Overall motivation for section 5.3 (Decoding task performance vs brain alignment)**
>
> * While the previous analyses revealed the importance of linguistic properties for brain alignment at individual layers, the main motivation for Section 5.3 is to understand the contribution of each linguistic property to the trend of brain alignment across layers.
> * This section provides more quantitative evidence and qualitative analyses by measuring the correlation across layers between the differences in decoding task performance from pretrained and residual BERT and the differences in brain alignment of pretrained BERT and residual BERT.
> * We also look at the correlation between the drops in performance in decoding and encoding that are due to the removal of a specific linguistic property. Therefore this evidence is more direct.
>
> * Overall, Section 5.3 helps us understand the contribution of each linguistic property to the trend of brain alignment across layers. It provides multiple neuroscience insights at the whole brain level, ROI level and sub-ROI level of language network.
>
> **2. The layer-wise decoding results are not so precise in terms of what ling property maps to what layer.  Is there any way to estimate the variance of these terms or find a definition for when something is robustly represented more in some than other layers? Can we do more to understand this?**
>
> * As the reviewer points out, we observe that a linguistic property doesn’t precisely map to any particular layer (Fig 7 and Fig 13 in the Appendix).
> * Instead, we focus on the brain alignment trend across all layers and specifically investigate the effect of the linguistic property on this trend, by measuring the correlation between the actual drop in performance resulting from the removal of a linguistic property and the subsequent reduction in brain alignment.
>
> **3. Removing any linguistic property results in a correlation drop from 0.1 to about 0.75. However, even this correlation is at about 0.75 when one removes all linguistic properties. Do you have an explanation why this might be?**
>
> * Fig 3a in the paper reports the average Pearson correlation across all layers of pretrained BERT and all voxels (i.e. whole brain analysis).
> * It is possible that we don't see much difference on average because different linguistic properties may be related to different parts of the network and different brain regions.
> * To investigate this possibility, we created a new brain plot (*Fig 2 in the rebuttal PDF*) by measuring the correlation across layers between voxelwise brain alignments before and after removing all linguistic properties. This shows that removing all properties shows very different region-level brain maps compared to removal of individual linguistic properties (Fig 4 in paper), and not much substantial brain alignment is left in the key language regions after removing all linguistic properties.
>
> * Please check the *rebuttal PDF* at “Common responses”.

---

> > ### Comment · Reviewer_wwF4 · 2023-08-18
> >
> > I thank the authors for the thoughtful response. I especially appreciate the added experiment and the explanation behind the motivation of section 5.3. To me, the intended audience of the paper and the implications of the findings are still not quite addressed -- can the insights inform AI engineering, computational modeling in neuroscience, or is it another method for making models in general more explainable? From the introduction and abstract, the last of the three seems most likely to me but for that it's generally very detached from other model explainability work.
> > If the authors can provide additional clarification on this point, I'll revise my overall recommendation.

---

> > > ### Author Response · Authors · 2023-08-19
> > >
> > > Thank you for stressing this point. We agree that including a more in-depth explanation for the immediate and long-term intended impact of the work will strengthen the impact even more. The insights gained from our work could have implications for AI engineering, neuroscience and interpretability of models; some in the short-term others in the long term.
> > >
> > > **AI engineering:** Our work most immediately fits in with the neuro-AI research direction that specifically investigates the relationship between representations in the brain and representations learned by powerful neural network models. This direction has gained recent traction especially in the domain of language, thanks to advancements in language models (Schrimpf et al. 2021 PNAS, Goldstein et al. 2022 Nature Neuroscience). Our work most immediately contributes to this line of research by understanding the reasons for the observed similarity in more depth. Specifically, our work can guide linguistic feature selection, can facilitate improved transfer learning and help in development of cognitively plausible AI architectures.
> > >
> > > **Computational Modeling in Neuroscience:** Our work enables cognitive neuroscientists to have more control over using language models as model organisms of language processing.
> > >
> > > **Model Explainability:** In the longer-term, we hope that our approach can contribute to another line of work that uses brain signals to interpret the information contained by neural network models (Toneva and Wehbe, 2019 NeurIPS; Aw & Toneva, 2023 ICLR). We believe that the addition of linguistic features by our approach can further increase the model interpretability enabled by this line of work.
> > >
> > > *We will add this discussion to the revised manuscript.*

---

> > > > ### Author Response · Authors · 2023-08-21
> > > >
> > > > Dear Reviewer wwF4,
> > > >
> > > > We appreciate your feedback and effort you have invested in evaluating our work.
> > > >
> > > > In response to your insightful comments, we have addressed the issues you highlighted. We believe these revisions significantly contribute to the clarity and completeness of the paper. We kindly request you to verify our response and consider updating your evaluation based on the revisions made.
> > > >
> > > > Should you have any further questions or suggestions, we are ready to provide additional information or clarification as needed.
> > > >
> > > > Thanks for your help

---

> > > > > ### Comment · Reviewer_wwF4 · 2023-08-22
> > > > >
> > > > > I thank the authors for engaging with my framing concern. I increased my score.

---

### Author Rebuttal · Authors · 2023-08-09

*We thank the reviewers for their strong positive, insightful and valuable comments and suggestions which are crucial for further strengthening our manuscript.*

**Why choose BERT over other models? (reviewers YHJW and p9mD)**

* The primary rationale behind our choice to incorporate the BERT model comes from the extensive body of research that aims to delve into the extent to which diverse English language structures are embedded within Transformer-based encoders, a field often referred to as "BERTology" [1] (as highlighted by Jawahar et al., 2019, and Mohebbi et al., 2021). We will clarify this.

[1] Rogers A, Kovaleva O, Rumshisky A. A primer in BERTology: What we know about how BERT works. Transactions of the Association for Computational Linguistics. 2021 Jan 1;8:842-66.

**Results for a GPT2 model (reviewer YHJW)**

* Based on the reviewers’ suggestion, we now perform experiments with a new model - GPT2. We find the results to be similar to BERT (i.e. a rich hierarchy of linguistic signals: initial to middle layers encode surface information, middle layers encode syntax, middle to top layers encode semantics.)
* Table 1 in rebuttal PDF reports the result for each probing task, before and after removal of the linguistic property from pretrained GPT2. We verify that the removal of each linguistic property from GPT2 leads to reduced task performance across all layers, as expected.
* We also report the layer-wise performance for pretrained GPT2 before and after the removal of one representative linguistic property–TopConstituents in Fig 1 of rebuttal PDF. We observe that the brain alignment is reduced significantly across all layers after the removal of the linguistic property (indicated with red cross in the figure). We are working on completing the results for other linguistic properties, but don’t expect them to differ too much from those based on BERT, given that the linguistic properties map onto similar layers (Table 1 in rebuttal PDF).

**Evidence that representations are not unexpectedly compromised by the removal method: comparison of brain encoding results using BERT with random parameters. (reviewer 9Zaj)**

* Based on the reviewers’ suggestion, we now perform brain encoding experiments with random weights of BERT.
  - Fig 3 (in the rebuttal PDF) shows brain predictivity performance using BERT with random weights is significantly worse compared to any of the “after” removal of linguistic property from pretrained BERT model. This shows that residual representations (i.e. removing a linguistic property from pretrained BERT) are informative and meaningful compared to random weights of BERT.

---

### Decision · Program_Chairs · 2023-09-21

**Decision:**

Accept (poster)

**Comment:**

The paper provides an analysis of correlations between BERT representations and fMRI data when abstracting away different linguistic properties from the BERT representations. The authors provide an in-depth analysis of the ways in which elimination of word length, tree depth, top constituents, tense, subject number and object number have on the overall and layer-wise alignment of BERT with fMRI data from 18 people. They find that removing any of these linguistic properties from the BERT representations reduces fMRI data alignment across all layers in the BERT model. Furthermore, syntactic properties are most relevant for the BERT-brain alignment globally, but semantic properties are highly locally aligned with brain regions generally associated with semantic processing.

This is an interesting paper, presenting novel empirical results, which can inform both the development of LMs and shed light on some aspects of how linguistic information is encoded in the human brain (at least in terms of how it relates to the information learnt by the language model from linguistic data). The paper is well-written, and the analyses are thorough and comprehensive (including appropriate controls etc.) The rebuttal addressed the reviewer's questions and concerns very well, providing motivation for the use of the specific models and removal method, presenting results on GPT-2 and implementing additional baselines which further support their claims. All in all, this is a technically sound, interesting and relevant paper, which would inform and stimulate further research in the community.